# CRISPR-enhanced human adipocyte browning as cell therapy for metabolic disease

Emmanouela Tsagkaraki [1,2,7], Sarah M. Nicoloro [1,7], Tiffany DeSouza[1], Javier Solivan-Rivera[1], Anand Desai[1], Lawrence M. Lifshitz[1], Yuefei Shen[1], Mark Kelly [1], Adilson Guilherme [1], Felipe Henriques [1], Nadia Amrani[2], Raed Ibraheim [3], Tomas C. Rodriguez [3], Kevin Luk[4], Stacy Maitland[4], Randall H. Friedline [1], Lauren Tauer [1], Xiaodi Hu[1], Jason K. Kim [1,5], Scot A. Wolfe [4,6], Erik J. Sontheimer [1,3,6], Silvia Corvera [1✉] & Michael P. Czech [1✉]

Obesity and type 2 diabetes are associated with disturbances in insulin-regulated glucose and lipid fluxes and severe comorbidities including cardiovascular disease and steatohepatitis. Whole body metabolism is regulated by lipid-storing white adipocytes as well as "brown" and "brite/beige" adipocytes that express thermogenic uncoupling protein 1 (UCP1) and secrete factors favorable to metabolic health. Implantation of brown fat into obese mice improves glucose tolerance, but translation to humans has been stymied by low abundance of primary human beige adipocytes. Here we apply methods to greatly expand human adipocyte progenitors from small samples of human subcutaneous adipose tissue and then disrupt the thermogenic suppressor gene NRIP1 by CRISPR. Ribonucleoprotein consisting of Cas9 and sgRNA delivered ex vivo are fully degraded by the human cells following high efficiency NRIP1 depletion without detectable off-target editing. Implantation of such CRISPR-enhanced human or mouse brown-like adipocytes into high fat diet fed mice decreases adiposity and liver triglycerides while enhancing glucose tolerance compared to implantation with unmodified adipocytes. These findings advance a therapeutic strategy to improve metabolic homeostasis through CRISPR-based genetic enhancement of human adipocytes without exposing the recipient to immunogenic Cas9 or delivery vectors.

[1] Program in Molecular Medicine, University of Massachusetts Medical School, Worcester, MA 01605, USA. [2] University of Crete School of Medicine, Crete 71003, Greece. [3] RNA Therapeutics Institute, University of Massachusetts Medical School, Worcester, MA 01605, USA. [4] Department of Molecular, Cell and Cancer Biology, University of Massachusetts Medical School, Worcester, MA 01605, USA. [5] Division of Endocrinology, Metabolism and Diabetes, Department of Medicine, University of Massachusetts Medical School, Worcester, MA 01605, USA. [6] Li Weibo Institute for Rare Diseases Research, University of Massachusetts Medical School, Worcester, MA 01605, USA. [7] These authors contributed equally: Emmanouela Tsagkaraki, Sarah M. Nicoloro.
✉email: Silvia.Corvera@umassmed.edu; Michael.Czech@umassmed.edu

The use of human cells as therapeutics offers major advantages over small molecule drugs and biologics in treating certain diseases based on their abilities to home to specific organs or cell types, initiate cell–cell interactions and secrete multiple bioactive factors[1,2]. Although still in early stages of development, cellular therapies have already had major impact on treatment of certain forms of cancer such as leukemia, lymphoma, melanoma and small cell lung carcinoma[3,4]. This approach involves genetic modification ex vivo of immune cells taken from a human subject to enhance their ability to disrupt malignancies upon infusion back into the same subject. In theory, this strategy should be effective in diseases in which cells with relevant therapeutic potential can be genetically modified to enhance that potential. For example, obesity and type 2 diabetes (T2D) involve multiple cell types that are disrupted in metabolic pathways in which their repair would alleviate disease[5–7]. Adipocytes represent such cells in which some of the defects have been identified[8–10]. Here we take advantage of recent discoveries revealing the utility of thermogenic adipocytes to function as major beneficial regulators of whole body metabolism in such metabolic diseases as T2D and obesity[11–14]. Thermogenic adipocytes, denoted as brown[15], beige[16] or brite[15,17], are distinct from the more abundant lipid storing white adipocytes not only by their high oxidative capacity and expression of mitochondrial uncoupling protein (UCP1), but also by their secretion of factors that enhance energy metabolism and energy expenditure[11–14]. One example is the secretion from brown and beige adipose tissue of neuroregulin 4 (NRG4) which has been shown to decrease hepatic lipid accumulation and improve glucose homeostasis in mice[18,19].

Multiple studies have demonstrated that implantation of mouse brown adipose tissue into obese, glucose intolerant mice can improve glucose tolerance and insulin sensitivity[20–22]. However, a bottleneck in taking advantage of this approach for therapeutic strategies in humans has been the scarcity of human beige adipocytes. Recently, human white and beige adipocytes expanded ex vivo from small samples of subcutaneous adipose tissue were shown to form robust thermogenic adipose tissue depots upon implantation into immune-compromised obese mice and to lower blood glucose levels[23]. Collectively, these data provide the framework to now apply genetic modifications to human adipocytes to further improve their therapeutic potential. The advent of clustered regularly interspaced short palindromic repeats (CRISPR) methods have greatly advanced progress towards enhanced cell therapy approaches[24,25]. Such genetic modifications must be performed by methods that minimize off-target effects and eliminate the presence of foreign reagent proteins in adipocytes prior to their implantation, which are major goals of our present studies.

The Nrip1 gene is an attractive target to enhance the therapeutic potential of adipocytes in obesity and diabetes. NRIP1 protein (also denoted as RIP140) had been shown to strongly suppress glucose transport, fatty acid oxidation, mitochondrial respiration, UCP1 expression as well as secretion of such metabolically beneficial factors including NRG4[26–28]. NRIP1 protein functions as a transcriptional co-repressor that attenuates activity of multiple nuclear receptors involved in energy metabolism, including estrogen related receptor (ERRα), peroxisome proliferator activated receptor (PPARγ) and thyroid hormone receptor (TH)[29]. NRIP1 knockout in white adipocytes upregulates genes that are highly expressed in brown adipocytes, enhancing glucose and fatty acid utilization and generating heat. Nrip1 ablation in mice elicits a lean phenotype under high fat diet conditions, and greatly enhances energy expenditure, glucose tolerance and insulin sensitivity[26]. However, NRIP1 is not an attractive target for conventional pharmacological intervention as

it is not an enzyme, and it also has a multiplicity of tissue specific roles such as regulating the estrogen receptor in the reproductive tract[29]. Thus, targeting NRIP1 selectively within adipocytes ex vivo represents an ideal approach to capture its therapeutic potential without undesirable side effects. Here we show high efficiency disruption of NRIP1 in mouse and human progenitor cells ex vivo by ribonucleoproteins (RNPs) of Cas9 protein and single-guide RNA (sgRNA) prior to their differentiation into adipocytes to enhance their beige characteristics and improve their therapeutic activities following implantation into obese mice.

## Results

**SpyCas9/sgRNA RNPs for ex vivo gene editing.** In order to define optimal sites for disruption of the Nrip1 gene in mouse preadipocytes, we designed 7 sgRNAs targeting various locations of the mouse genomic DNA in exon 4, which contains the entire open reading frame (Fig. 1a). A key aspect of our strategy in targeting the Nrip1 gene was to employ methods that would ablate its expression in adipocytes but not cause immune responses upon implantation of the cells. CRISPR-based methods eliciting continuous expression of Cas9/sgRNA to modify adipocytes that function in vivo have been reported[30,31], but they expose recipients to Cas9 and delivery agents that cause immune responses[32]. Direct administration of Cas9/sgRNA complexes in mice have not been adipocyte-specific and could cause undesirable effects in other tissues[30,33]. RNP complexes of SpyCas9/sgRNA are desirable vehicles for such modifications since they are rapidly degraded following DNA disruption[34]. A previous attempt at delivery of such CRISPR-based complexes to adipocytes were suboptimal as efficiencies of delivery of RNPs to these cells was only modest[28]. We overcame these deficiencies by disrupting Nrip1 in mouse preadipocytes with RNP complexes of Cas9 and sgRNA by modifying electroporation methods[35] described for other cell types (Supplementary Fig. 1) and confirmed Cas9 protein is rapidly degraded following indel formation in preadipocytes using RNPs containing sgRNA M6 (Fig. 1b). Electroporation conditions were developed to optimize the efficiency of Nrip1 gene targeting in mouse preadipocytes by Cas9/sgRNA RNPs without perturbing their differentiation into adipocytes (Fig. 1d and Supplementary Fig. 1). Thus, the efficiencies of indel formation by the 7 different sgRNAs against various regions of Nrip1 gene exon 4 (Fig. 1a) were uniformly sustained in the 90% range in preadipocytes and upon their differentiation into adipocytes (Fig. 1d). Indels were quantified by Sanger sequencing data analysis of PCR fragments spanning the upstream and downstream double stranded breaks of the Nrip1 genomic DNA (Fig. 1c, e) with little change in the total Nrip1 mRNAs (Fig. 1f). High frequencies of frameshift mutations in Nrip1 by all 7 sgRNAs were observed and similar indels were found in the corresponding Nrip1 mRNA species, as exemplified by sgRNA M3 and M4 (Fig. 1e, Supplementary Fig. 2a, b).

While the mRNA of Nrip1 was equally abundant in all groups, indicating little or no increased degradation due to disruption (Fig. 1f), surprisingly, not all of the sgRNAs were effective in eliciting loss of the NRIP1 protein (Fig. 1g). Consistent with these data, thermogenic responses to the various sgRNAs as reflected by elevated expression of UCP1 mRNA (Fig. 1h) and protein (Fig. 1i) correlated with the loss of native full length NRIP1 protein. Taken together, these data show that sgRNAs targeting the regions of Nrip1 DNA that encode the N-terminal region of the NRIP1 protein are not effective in eliminating synthesis of functional NRIP1 protein. Most likely, additional transcription or translation start sites beyond these target sites are functional under these conditions (Supplementary Fig. 2c). Thus, sgRNAs

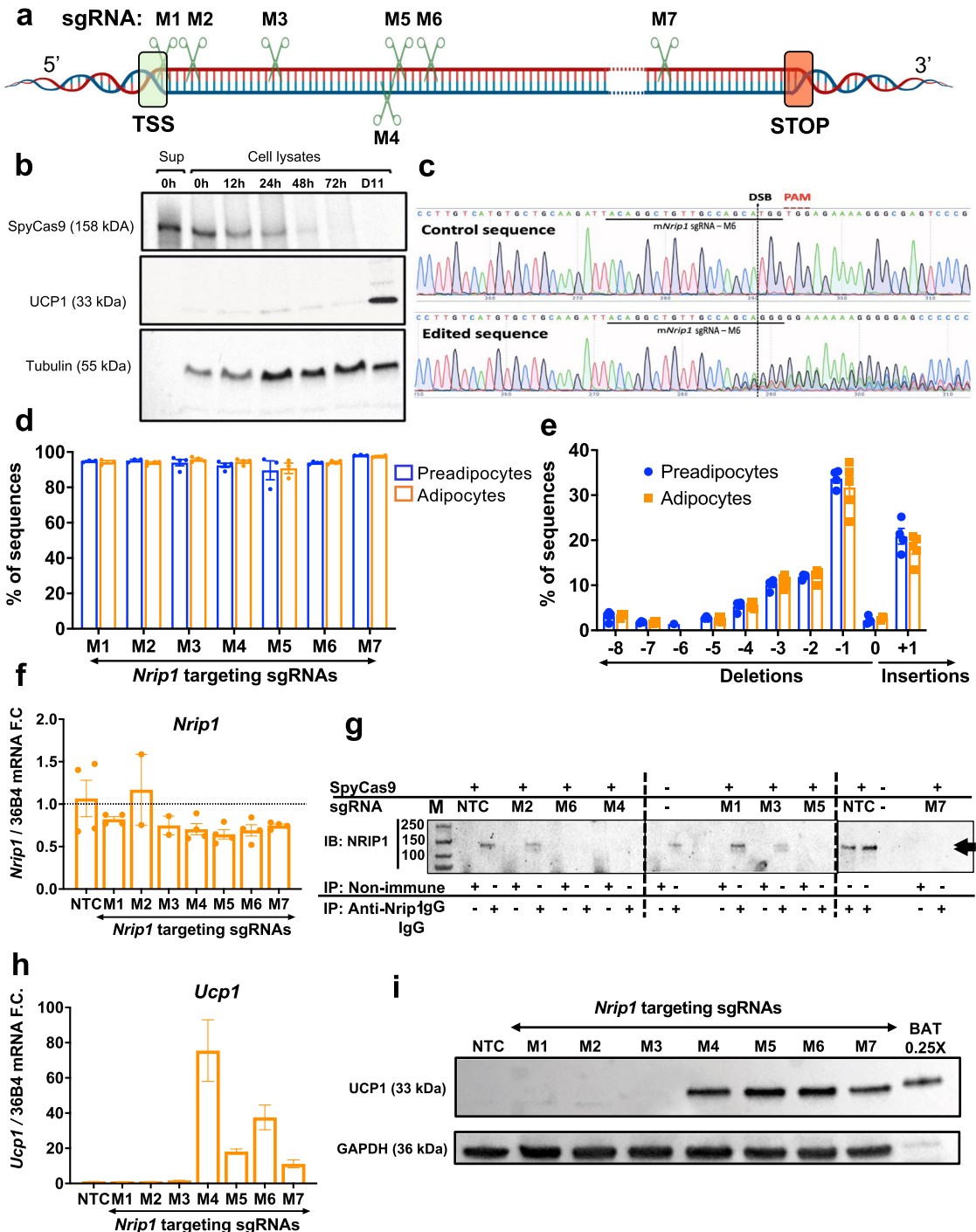

that are optimal for inducing thermogenic genes must be identified by such screening methods.

**Thermogenic activity in NRIP1KO adipocytes from male and female mice.** Experiments as described in Fig. 1 were performed on preadipocytes from both male and female mice to determine effects of *Nrip1* disruption on adipocyte thermogenesis with equally high editing efficiency. Upregulation of *Ucp1* mRNA was over 100 fold in *Nrip1* deficient adipocytes derived from both male and female mice (Fig. 2a), and the fold changes in expression of many other genes were also nearly identical in these genetically modified adipocytes from the different sexes (Fig. 2b). This similarity in response to *Nrip1* disruption in male versus

female mice was also observed in the upregulation of UCP1 protein expression (Fig. 2c).

In addition to UCP1 upregulation, NRIP1KO adipocytes also exhibited higher expression of mitochondrial respiratory chain components UQCRC2 of complex III and SDHB of complex II (Fig. 2d). Together with the increased UCP1 expression, the upregulation of these and many other known metabolic enzymes combine to dramatically affect the respiration rates and characteristics of NRIP1KO adipocytes. Confirming this expectation, we found that while there is little change in oxygen consumption between control and NRIP1KO adipocytes under basal (Fig. 2e, Supplementary Fig. 3), unstimulated conditions, norepinephrine (NE) much more robustly increased NRIP1KO adipocyte respiration (Fig. 2f, Supplementary Fig. 3) due to its

**Fig. 1 High efficiency *Nrip1* gene disruption at 7 loci by SpyCas9/sgRNA RNPs produces variable degrees of NRIP1 protein loss and UCP1 upregulation in murine primary adipocytes. a** Mapping of the sgRNAs M1-M7 targeting various loci of murine *Nrip1* coding region which is entirely located in exon 4 (TSS transcription start site, STOP stop codon). **b** Time-course of SpyCas9 protein degradation detected by Western Blotting in cell lysates at various time points (0–72 h and day 11 after transfection, which is day 6 of differentiation) after electroporation with RNPs of SpyCas9:sgRNA (3:4 µM). Sup denoted supernatant containing SpyCas9 at 0 h. **c** Sanger sequencing traces of control vs *Nrip1* disrupted cells with sgRNA-M6 showing the sgRNA binding site (solid black line), PAM (red), the double-strand-break (denoted as DSB) site (dashed black line) on the sgRNA-M6 targeting locus and the traces downstream of the DSB created by the DNA repair mechanisms (figure created with SnapGene). **d** Editing efficiency as evaluated with indel percentage 72 h after the transfection of primary preadipocytes (blue) and differentiation to mature primary adipocytes (orange). **e** Indel distribution of *Nrip1* sgRNA-M6 with frameshift indels that are sustained after differentiation. **f** *Nrip1* gene expression detected by RT-PCR in mature adipocytes targeted with the different sgRNAs. **g** Immunoprecipitation assay for NRIP1 (140 kDa, arrows at right) in mature primary adipocytes on day 6 post differentiation targeted with the different sgRNAs. The total lysate protein amount used in the assay was 250 µg per sample. M denotes molecular weight marker. Dashed lines separate different gels. **h** *Ucp1* expression by RT-PCR in mature adipocytes targeted with the different sgRNAs compared to non-targeted control cells. **i** Western blot for UCP1 protein (33 kDa) in mature adipocytes on day 6 post differentiation targeted with the different sgRNAs. Lanes 1–8 were loaded with 20 µg of total protein while lane 9 was loaded with 5 µg of total protein isolated from mouse BAT. NTC non-targeting control. In (**d**, **e**, **f** and **h**) bars denote mean and error bars denote Mean ± S.E.M. $n \geq 3$ biologically independent replicates. Detailed *n* per condition and molecular weight markers in (**b**) and (**i**) are shown in the source data.

ability to activate UCP1-mediated uncoupling of mitochondria. This concept was verified upon addition of oligomycin, an inhibitor of coupled but not uncoupled respiration, which was much less effective in inhibiting oxygen consumption in NE stimulated NRIP1KO adipocytes compared to its marked suppression of NE stimulated control adipocyte respiration (Fig. 2f, g, Supplementary Fig. 3). These data obtained with adipocytes derived from male mice were similar to the results observed with adipocytes from female mice (Fig. 2h–j). Altogether, the data in Fig. 2 and Supplementary Fig. 3 demonstrate that NRIP1 deficiency has similar effects to increase uncoupling and thermogenic properties of mitochondria from female and male adipocytes.

**Implantation of CRISPR-modified mouse adipocytes**. To test the ability of NRIP1-deficient adipocytes to improve metabolism in mice, large numbers of primary preadipocytes obtained from 2 to 3 week old mice were electroporated with RNPs consisting of either SpyCas9/non-targeting control (NTC) sgRNA or SpyCas9/sgRNA-M6 complexes (NRIP1KO), and then differentiated into adipocytes and implanted into wild type mice. The implanted mice were kept on normal diet for 6 weeks during the development of adipose tissue depots from the injected adipocytes, then placed on a high fat diet (HFD) regimen to enhance weight gain (Fig. 3a). Adipocytes treated with Cas9/sgRNA-M6 displayed upregulation of *Ucp1* and other genes highly expressed in brown adipocytes (e.g., *Cidea*) prior to transplantation (Supplementary Fig. 4a). A transient decrease in overall body weights were detected between mice implanted with RNPs containing the Cas9/sgRNA-M6 versus the Cas9/NTC-sgRNA group, but by 9 weeks of HFD no significant difference was observed (Fig. 3b). Nonetheless, implantation of NRIP1KO adipocytes prevented the increase in fasting blood glucose concentration due to HFD that occurs in the NTC adipocyte-implanted mice (Fig. 3c). Glucose tolerance was also significantly improved by implantation of NRIP1KO adipocytes (Fig. 3d, e). The implanted adipose tissue depots retained their elevated expression of UCP1 16 weeks after implantation, at which time they were excised for analysis (Fig. 4h). The livers and inguinal white adipose tissues (iWAT) from the Cas9/sgRNA-M6 group of mice had lower weights (Fig. 3f) and lower iWAT to body weight ratios (Supplementary Fig. 4c), revealing a strong systemic effect.

Livers of the NRIP1 deficient adipocyte-implanted mice were less pale (Fig. 3g), were smaller as assessed by lower liver to body weight ratios (Fig. 3h) and displayed lower expression of genes associated with free fatty acid uptake (*Cd36*) and with inflammation (*Mcp1*, *Tnfα*, *Il1β*) (Fig. 3i) compared to mice implanted with control adipocytes. Lipid droplets in the livers of

mice with implants of Cas9/sgRNA-M6-treated adipocytes (Fig. 3j) were greatly decreased as assessed by quantification of lipid droplet area (Fig. 3k), number and size (Supplementary Data Fig. 4j, k) as well as liver triglyceride determination (Fig. 3l). The decrease in hepatic lipid accumulation and inflammation in response to implantation of the NRIP1 depleted adipocytes suggests that this therapeutic approach might mitigate these T2D co-morbidities in humans[36].

Consistent with our finding that Cas9 was degraded prior to implantation of the engineered cells (Fig. 1b), the recipient mice had no signs of disease or distress. The excised implants had no macroscopic signs of inflammation or necrotic tissue and the *Mcp1* gene which signifies an inflammatory response was not upregulated in NTC or NRIP1KO implants compared to the endogenous iWAT tissue (Supplementary Fig. 4i). Additional assessment of potential inflammatory responses showed no difference in plasma levels of IL-1β, IL-6, IL-10, MCP-1 or TNFα in the NTC or NRIP1KO mice compared to untreated control mice, while LPS treated mice serving as positive controls displayed greatly increased levels of these factors (Supplementary Fig. 4l).

**Gene expression profiles of NRIP1KO mouse adipocytes and implants**. To further characterize the NRIP1KO mouse adipocytes in an unbiased manner, transcriptome analysis was performed by RNA-sequencing of mouse primary mature adipocytes on day 6 of adipogenic differentiation of either non-treated control adipocytes (no electroporation or RNPs, NT) or SpyCas9 + sgRNA-NTC treated adipocytes or adipocytes treated with either SpyCas9 + sgRNA-M6 or SpyCas9 + sgRNA-M4 that target and deplete NRIP1 protein (NRIP1KO). Principal component analysis showed a distinct clustering of the control groups and the NRIP1KO samples independent of the specific sgRNA used, indicating distinct gene expression profiling (Fig. 4a). Differential gene expression comparison between all the control samples and the NRIP1KO samples revealed 902 significantly upregulated genes including *Ucp1, Cidea, Fabp3* and the secreted neurotrophic factor *Nrg4* (Fig. 4b) and 511 significantly down-regulated genes. Interestingly, the top upregulated pathways upon *Nrip1* disruption are related to mitochondrial respiration and fatty acid oxidation (Fig. 4c), while a detailed pathway analysis of the upregulated genes revealed functions including adaptive, diet-induced and cold-induced thermogenesis (Fig. 4d). In particular, 78 of the upregulated genes are associated with the cellular respiration pathway (Fig. 4e) and 18 of the upregulated genes are associated with thermogenesis (Fig. 4f). In order to assess the overall thermogenic potential, the ProFAT computational tool was applied to the three groups of samples (NT, NTC,

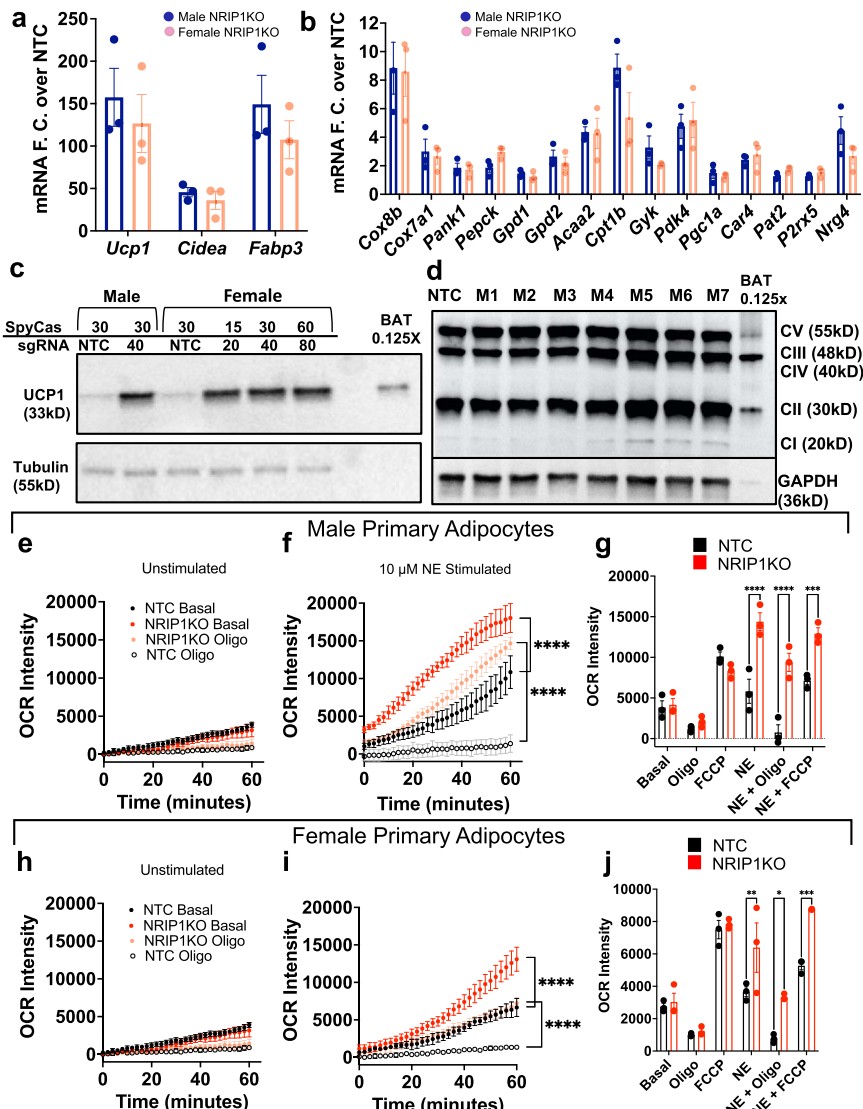

**Fig. 2 Primary mouse NRIP1KO adipocytes from both male and female mice display increased thermogenic gene expression and uncoupled oxygen consumption. a**, **b** *Ucp1*, *Cidea*, *Fabp3* and thermogenic gene expression, fold change over NTC. In (**a**, **b**) male = blue ($n = 3$), female = peach ($n = 3$). Error bars denote mean ± SEM. **c** UCP1 (33 kDa) protein expression in primary adipocytes, 6 days post differentiation with various RNP concentrations. Lanes 1 to 6 contain 20 μg of protein lysate and lane 8 contains 2.5 μg of protein lysate from mouse BAT. Tubulin, as loading control. **d** Oxidative phosphorylation protein expression in primary male adipocytes on day 7 post differentiation targeted with NTC or NRIP1 sgRNA-M1-M7. Lanes 1 to 8 contain 20 μg protein while lane 9 contains 2.5 μg protein (mouse BAT). GAPDH, loading control. **e** Oxygen consumption rates (OCR) in mature primary male adipocytes targeted with NTC or NRIP1 sgRNA-M6 without norepinephrine (NE) stimulation and (**f**) with NE stimulation. Statistical comparison for OCR in technical replicates was done for (**e**, **f**) using one-way ANOVA with Sidak's multiple comparison test. ****$p < 0.0001$ (**g**). Summary of OCR in panels (**e**, **f**) at 40 min in primary male adipocytes with and without NE stimulation. Statistical comparison using two – way ANOVA with Sidak's multiple comparison test. ****$p < 0.0001$, ***$p = 0.0003$. **h** OCR in mature primary female adipocytes targeted with NTC or NRIP1 sgRNA-M6 without NE stimulation and (**i**) with NE stimulation. Statistical comparison for OCR technical replicates for (**h**, **i**) was using one-way ANOVA with Sidak's multiple comparison test. ****$p < 0.0001$ (**j**). Summary of OCR at 40 min in primary female adipocytes. Statistical comparison using two – way ANOVA with Sidak's multiple comparison test. NE **$p = 0.0065$, NE + Oligo *$p = 0.0112$, NE + FCCP ***$p = 0.0005$. Biological replicates confirmed the increased OCR by NE treated NripKO cells compared to NE treated NTC cells ($p = 0.02$, $n = 4$) as well as increased OCR by NE treated NripKO cells plus oligomycin compared to NE plus oligomycin in NTC cells ($p = 0.03$, $n = 4$) (Supplementary Fig. 3). Molecular weight markers in (**c** and **d**) are shown in the source data. Error bars in (**e–j**) denote mean ± SEM.

NRIP1KO), indicating >90% "brown" adipocyte type probability in the NRIP1KO cells as opposed to the NTC and NT samples that ranged from 0% to only 20% brown probability (Fig. 4g)[37].

In order to examine the extent to which the altered transcriptome by NRIP1KO is sustained by the end of the study, the top 15 upregulated genes with >10,000 normalized reads in all NRIP1KO adipocyte samples were selected for screening by RT-PCR in the mouse implants 16 weeks after implantation (Fig. 4h). All 15 genes that were screened, tended to be more highly

expressed in the NRIP1KO implant samples than in the control implant tissue, with some reaching significance despite the 4-month time lapse in vivo and the potential presence of endogenous cells that had migrated into the implants (Fig. 4h). In agreement with the in vitro studies, the mouse NRIP1KO adipocytes possessed a "brown" profile demonstrated by increased thermogenic genes and an increase in the expression of mitochondrial respiration components that was sustained through implantation and many weeks of HFD feeding.

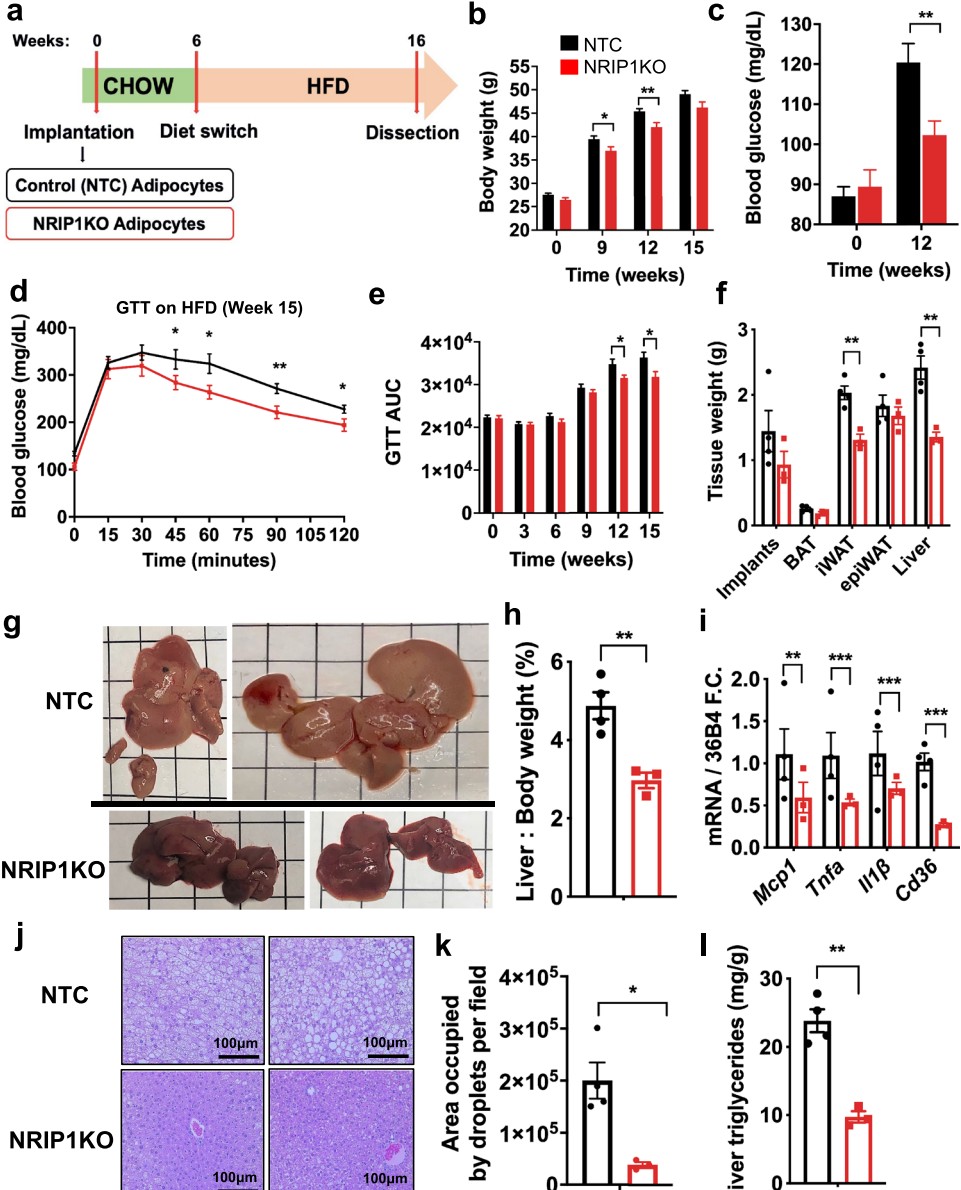

**Fig. 3 Implantation of NRIP1-depleted mouse adipocytes improves glucose tolerance and markedly decreases iWAT weight and liver triglyceride accumulation in recipient mice.** Mice were implanted with either mouse adipocytes previously transfected with NTC sgRNA/Cas9 RNPs or with sgRNA-M6/Cas9 RNPs (NRIP1KO). **a** Schematic protocol of implantation of murine NTC adipocytes or NRIP1KO adipocytes into C57BL/6 wild type mice followed by 60% kCal fat diet (HFD). **b** Total body weights of recipients on chow before implantation and 6, 9, 12, 15 weeks after implantation on HFD. P values *9 weeks = 0.034; **12 weeks = 0.008; 15 weeks = 0.058 **c** Fasting blood glucose concentrations at baseline (p = 0.626) and 12 weeks (p = 0.005) post implantation (6 weeks on HFD). **d** Glucose tolerance test (GTT) after 16-hour overnight fasting in implant recipients after 9 weeks on HFD. P values *45 min = 0.015; *60 min = 0.011; **90 min = 0.004; *120 min = 0.022 **e** Bar graphs of areas under the curve from GTTs in implant recipient mice on chow and after 9, 12 and 15 weeks on HFD. P values *12 weeks = 0.022; *15 weeks = 0.016. **f** Weight of total bilateral implants, whole BAT, total bilateral iWAT, epiWAT and total liver as measured after dissection. P values **iWAT = 0.004; **Liver = 0.005. **g** Macroscopic images of the whole livers of the implant recipients after dissection (square = 1 cm²). **h** Liver over whole body weight percentage. p = 0.008. **i** Expression of genes related to inflammation and hepatic steatosis in the livers of implant recipients detected by RT-PCR. P values *Mcp1 = 0.02; ***Tnfa = 0.00001; **Il1b = 0.001; ****Cd36 < 0.000001. **j** Hematoxylin and eosin (H&E) stain on liver histology of the implant recipients at 20X magnification. **k** Quantification of total H&E images of implant recipients' livers for total area occupied by lipid droplets per field (p = 0.011). **l** Triglyceride measurements in pulverized liver extracts after dissection. (p = 0.001). black = NTC adipocyte implant recipients, red = NRIP1KO adipocyte implant recipients; in **b–e**: NTC (n = 13); NRIP1KO (n = 14), in **f–l**: NTC (n = 4); NRIP1KO (n = 3). Each n represents number of biologically independent mice. Bars represent the mean. Error bars denote mean ± SEM. *p < 0.05, **p < 0.01, ***p < 0.001 by unpaired two-tailed T test.

**Translation to human adipocytes.** To translate these CRISPR-based methods to human adipocytes, adipocyte progenitors were obtained from small samples of excised subcutaneous adipose tissue as previously described[23]. Electroporation conditions were tested to optimize efficiency of indel formation using various

sgRNAs directed against regions of the *NRIP1* exon 4 at locations roughly similar to those we targeted in the mouse genomic DNA (compare Fig. 1a to Fig. 5a). Efficiencies of *NRIP1* gene disruption were observed with several sgRNAs in the 90% range (Fig. 5b), with indel distributions very similar in preadipocytes and

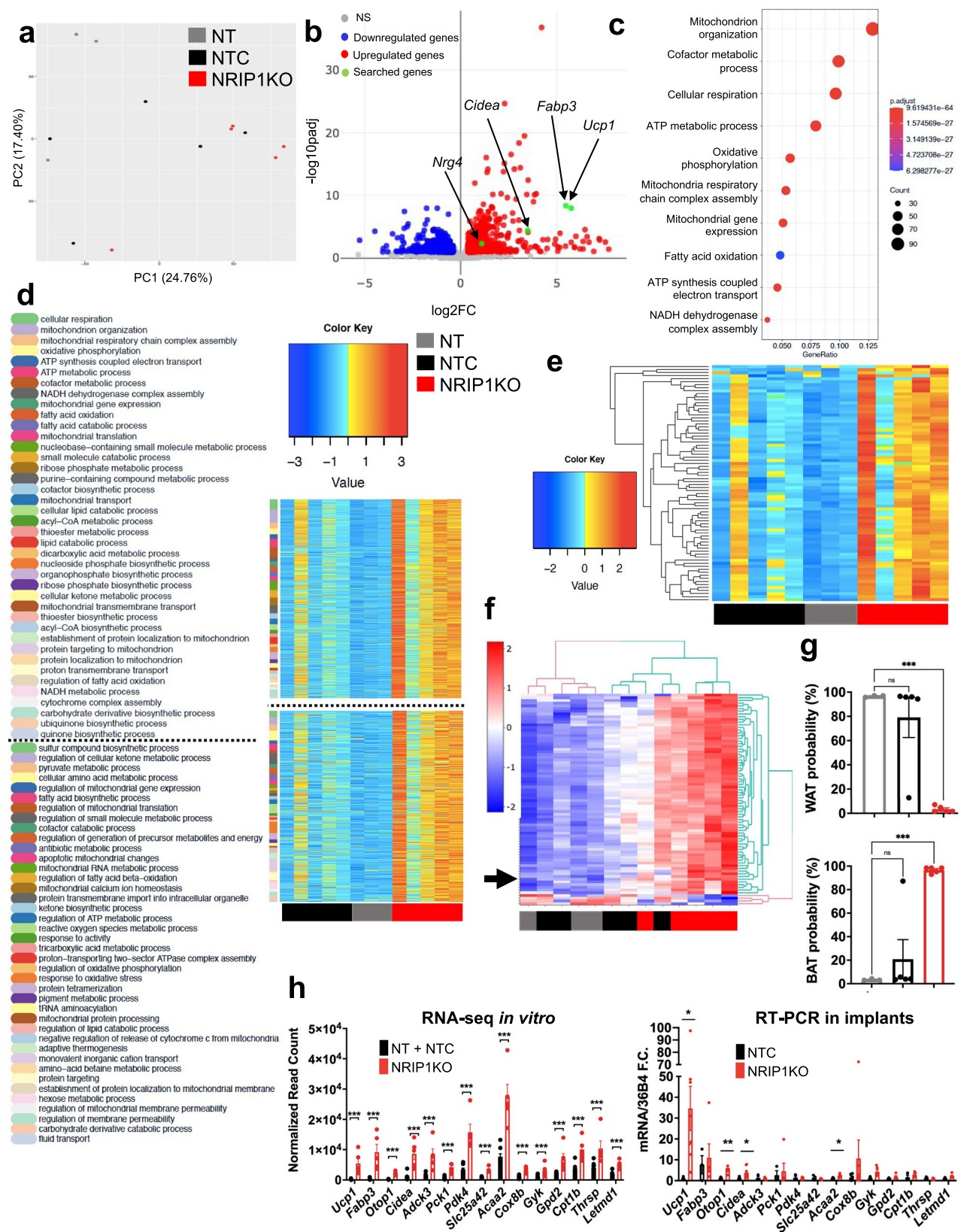

adipocytes (Fig. 5c). Electroporated, NRIP1 deficient human preadipocytes could be readily differentiated to adipocytes without apparent disruption following indel formation (Fig. 5d). *NRIP1* mRNA was equally abundant in all conditions (Fig. 5e). Similar to the findings in mouse adipocytes, NRIP1 protein was not depleted when using an sgRNA targeting an upstream site in

the genomic DNA, as shown with sgRNA-H1 (Fig. 5f). However, with sgRNA-H5, significant depletion of NRIP1 protein was observed (Fig. 5f) and associated with an 80-fold upregulation of *UCP1* expression (Fig. 5g, h). Functional NRIP1 protein can still be expressed despite high efficiency indel formation in the N-terminal NRIP1 encoded region of the human *NRIP1* gene,

**Fig. 4 RNA sequencing on non-treated, NTC and NRIP1KO mature adipocytes on day 6 after differentiation. a** Principal component plot prior to DEseq analysis. **b** Volcano plot of upregulated (red) and downregulated (blue) genes between control unedited adipocytes (NT and NTC) and NRIP1KO adipocytes with highlighted (green) Ucp1, Cidea, Fabp3, Nrg4. **c** Top 10 pathways associated with all upregulated genes detected. **d** Heatmap of pathway analysis of upregulated genes ranked according to the p value. **e** Heatmap of the 78 genes associated with cellular respiration that were upregulated in NRIP1KO adipocytes. **f** Heatmap of genes related to thermogenesis. Arrow shows Ucp1. **g** Browning probability calculated using the ProFAT online tool. $p = 0.0004$ by One-way ANOVA. **h** Left: top 15 upregulated genes by RNA sequencing where all NRIP1KO samples have >1000 normalized reads and padj > 0.1 (Black = NTC + NT) and NT, (Red = NRIP1KO). Right: Screening of these 15 genes in the excised implant NTC (black) or NRIP1KO (red) tissue by RT-PCR. P values *Ucp1 = 0.011; **Otop1 = 0.0017; *Cidea = 0.013; *Acaa2 = 0.018 by unpaired two-tailed T-test. Cut-offs were set at padj < 0.1 and fold change > 1.3. NT (n = 3), NTC (n = 5), NRIP1KO (n = 5). NRIP1KO adipocytes were transfected with SpyCas9 and either Nrip1 sgRNA-M4 or sgRNA-M6.

similar to what was observed with mouse (Fig. 1g). Interestingly, *NRIP1* disruption in combination with the adenylate cyclase activator forskolin synergistically increased *Ucp1* expression (Fig. 5i), consistent with the concept that the cAMP pathway acts independently but coordinately with NRIP1 function.

**Gene expression profiles of NRIP1KO human adipocytes**. To characterize the effect of disrupting *NRIP1* in human adipocytes, RNA-seq was performed on NTC or NRIP1KO human mature adipocytes on day 7 after adipogenic differentiation. Interestingly, most differentially expressed genes were found to be upregulated and only 9 genes were downregulated (Fig. 6a). In agreement with the findings in mouse adipocytes, the top ten pathways associated with upregulated genes by NRIP1KO and the majority of all cellular functions revealed by pathway analysis are related to mitochondrial respiration and fatty acid oxidation (Fig. 6b–d) with 16 and 9 out of the upregulated genes being associated with these pathways, respectively. To calculate the thermogenic potential of the NRIP1KO human adipocytes, the ProFAT computational tool[37] was applied to the raw gene expression datasets of the two groups of samples (NTC vs NRIP1KO), indicating a much greater brown-like probability in the NRIP1KO cells (Fig. 6e). Many of the human genes known to be related to thermogenesis were found to be highly expressed in NRIP1KO cells, with *UCP1* ranked as the most upregulated gene overall (53-fold) (Fig. 6f).

**Implantation of CRISPR-modified human adipocytes**. To test the efficacy of NRIP1-depleted human adipocytes to provide metabolic benefits in obese glucose intolerant mice, immune-compromised NOD.Cg-Prkdc^scid Il2rg^tm1Wjl/SzJ (NSG) mice were utilized that lack T cells, B cells and natural killer cells and accept human cell implants in the protocol depicted (Fig. 7a, b). *NRIP1* gene disruption was about 80% with the sgRNA-H5 (Fig. 7c) and circulating human adiponectin (Fig. 7d) was the same from NTC vs sgRNA-H5 groups, indicating similar levels of adipose tissue formation. While no body weight difference between groups was detected on normal diet, a highly significant decrease in weight gain on the HFD was observed in mice implanted with *NRIP1* disrupted adipocytes (Fig. 7e). Importantly, mice with control human adipocyte implants displayed significantly decreased glucose tolerance 3 weeks after starting a HFD while animals with NRIP1-depleted adipocyte implants did not (Fig. 7f–i). The difference in glucose tolerance between the two groups at the end of the study was highly significant (Fig. 7h, i). Relative liver to body weight ratios (Fig. 7j) and liver triglycerides (Fig. 7k) were also decreased when implants were performed with NRIP1 deficient adipocytes compared to control adipocytes.

The genome-wide, unbiased identification of DSBs enabled by sequencing (GUIDE-Seq) was used to evaluate the potential off-target editing present in the mouse and human adipocytes described above[38]. In the mouse primary preadipocytes, a total of 12 potential off-target sites were detected and ranked by number

of reads and number of sgRNA mismatches in the edited cells (Supplementary Fig. 5a). A total of 11 off-target sites, mOT1-11, with up to 5 sgRNA mismatches were selected to assess editing with amplicon NGS. Out of these off-target sites screened, five off-target sites were found to have >1% editing with mOT1, mOT8, mOT9, mOT10 being located in unannotated genomic regions and the rest in intronic regions (Supplementary Fig. 5b). In the human progenitors, GUIDE-seq revealed zero detectable off-target sites in the edited cells compared to the NTC cells, even after transfecting the cells with higher amount of dsODN (Supplementary Fig. 5c, d). To further validate the human GUIDE-seq results, Cas-offinder[39] was used to select a total of 5 off-target sites which were screened by amplicon NGS, revealing no off-target editing (Supplementary Fig. 5e, f).

**Discussion**

A major goal of the present study was to advance the application of CRISPR technology to metabolic disease in the context of a potential therapeutic strategy. The starting point of our experimental plan was the success of many laboratories in demonstrating the efficacy of implanting mouse or human brown/beige adipocytes into glucose intolerant mice to alleviate diabetes[21,23,40–42]. Five key criteria were incorporated into our approach: 1. Generation of large numbers of adipocyte progenitors from small samples of human adipose tissue, 2. Identification of a strong suppressor of adipocyte beiging for targeting by CRISPR to optimize the therapeutic benefit of adipocyte implantation, 3. Stealth administration of SpyCas9/sgRNA to cells that would not expose recipient mice to immunogenic reagents, 4. Minimizing off target effects, and 5. High efficiency gene disruption in which most cells ex vivo are affected in a single step without a cell selection step. Taken together, the data presented here show that our methods using CRISPR-based RNPs targeting Nrip1 to a large extent satisfy the above criteria. These methods can indeed enhance browning of mouse or human adipocytes ex vivo at high efficiency without the use of expression vectors to improve glucose homeostasis in obese mice.

Targeting *Nrip1* in these studies proved highly effective, consistent with previous studies by other laboratories[26,29] and ours[27,28] indicating that NRIP1 is one of the most powerful suppressors of adipocyte beiging. Importantly, NRIP1 deletion in preadipocytes does not diminish differentiation of these cells into adipocytes. Although CRISPR-based upregulation of UCP1 alone in implanted adipocytes can improve metabolism in mice[31], targeting *NRIP1* has the advantage of upregulating expression of many genes that have favorable metabolic effects in addition to UCP1 as well as secreted factors such as NRG4. Moreover, NRIP1 disruption synergizes with the cAMP pathway (Fig. 5h), known in adipocytes to have broad effects on mitochondrial respiration and metabolism[43]. Since UCP1 expression in NRIP1KO adipocytes does not reach the level of mouse BAT, our approach can likely be improved by further enhancing adipocyte browning through disrupting combinations of targets in addition to NRIP1.

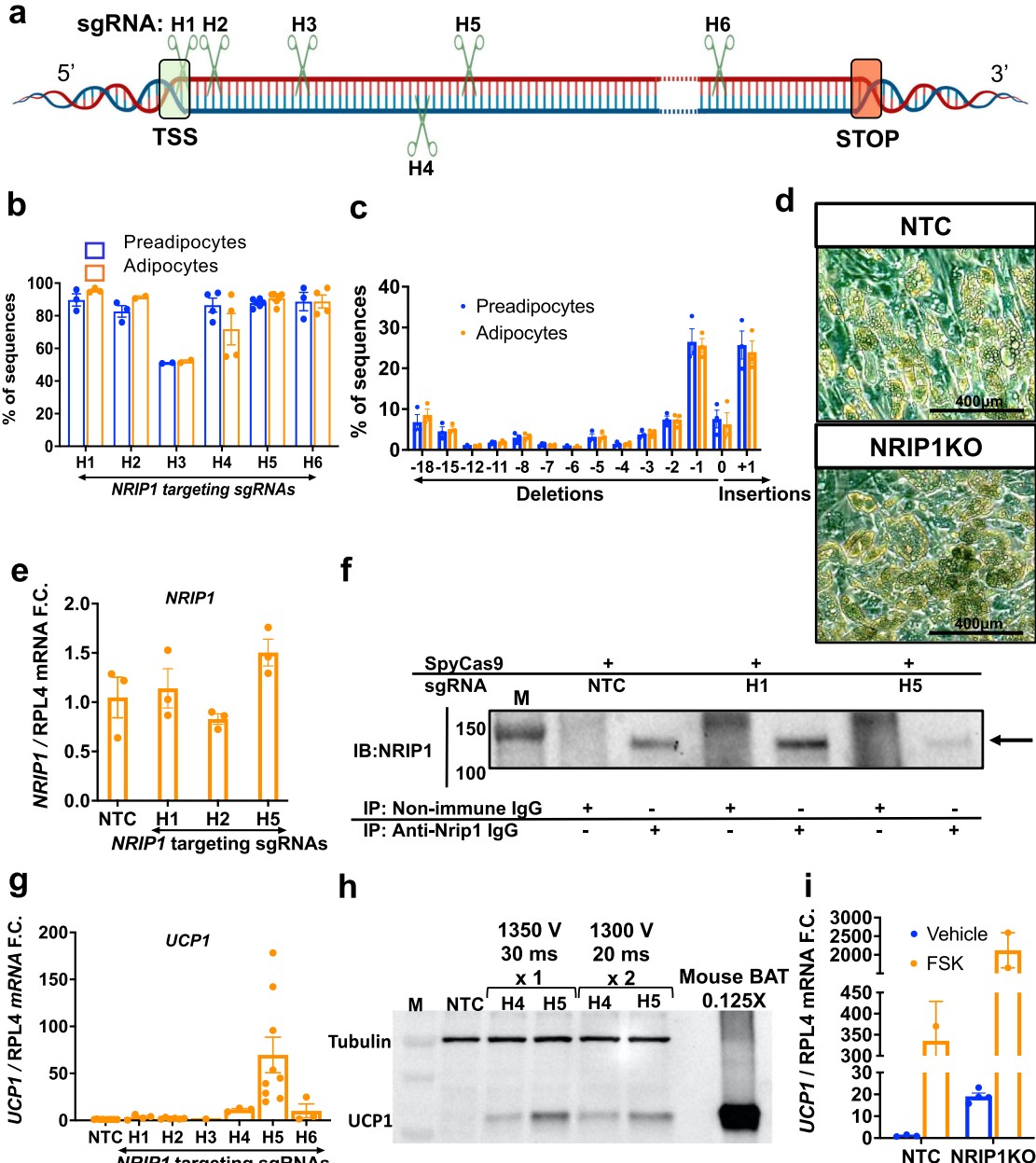

**Fig. 5 High efficiency *NRIP1* disruption in human adipocytes by SpyCas9 reveals variable UCP1 upregulation in a screen of sgRNAs targeting different loci of *NRIP1*. a** Mapping of sgRNAs H1-H6 screened against human *NRIP1* coding region entirely located in exon 4. **b** Editing efficiency as evaluated with indel percentage 72 h after the transfection of primary preadipocytes (blue) and differentiation to mature primary adipocytes (orange). **c** Indel distribution of *NRIP1* sgRNA-H5 with frameshift indels that are sustained after differentiation. **d** Microscopic image of cell culture of non-targeted control (top) and *NRIP1* disrupted (bottom) mature adipocyte morphology in cell culture at 10X magnification and scale bar represents 400 μm. **e** *NRIP1* gene expression by RT-PCR in mature adipocytes on day 7 post differentiation targeted with the different sgRNAs. **f** Immunoprecipitation assay for NRIP1 (140 kDa, arrows at right) in mature human adipocytes on day 7 post differentiation targeted with the different sgRNAs. The total protein lysate amount used in the assay was 250 μg per sample. M denotes molecular weight marker. **g** *UCP1* expression by RT-PCR in mature adipocytes on day 7 post differentiation targeted with the different sgRNAs compared to non-targeted control cells. **h** Western blot for UCP1 protein (33 kDa) in mature adipocytes on day 7 post differentiation targeted by the sgRNAs-H4 and -H5 with two different electroporation optimization protocols. Lanes 2–6 were loaded with 20 μg of total protein while lane 9 was loaded with 2.5 μg of total protein isolated from mouse BAT. **i** *UCP1* gene expression by RT-PCR in non-targeted control or NRIP1 depleted adipocytes on day 7 post differentiation after a 7-hour stimulation of forskolin 10 μM or vehicle. NTC non-targeting control. Bars denote mean. Error bars denote Mean ± S.E.M, $n \geq 3$ biological replicates. In (**f**, **h**) M denotes molecular weight marker.

The specific technical approach presented here also has several additional advantages, including the rapid electroporation procedure with minimal loss of viability of the transfected preadipocytes. Also, the brief exposure of cells to Cas9/sgRNA that we document here (Supplementary Fig. 1) reduces the potential for off target effects that are produced by long term expression of these reagents. The immunogenic SpyCas9 is readily degraded within the adipocytes prior to implantation (Fig. 1b), minimizing off-target effects (Supplementary Fig. 5). Indeed, we report here that while significant off-target indels were observed in the mouse preadipocytes, no significant off-target gene disruptions in the human progenitors were observed over background in a rigorous application of GUIDEseq and deep

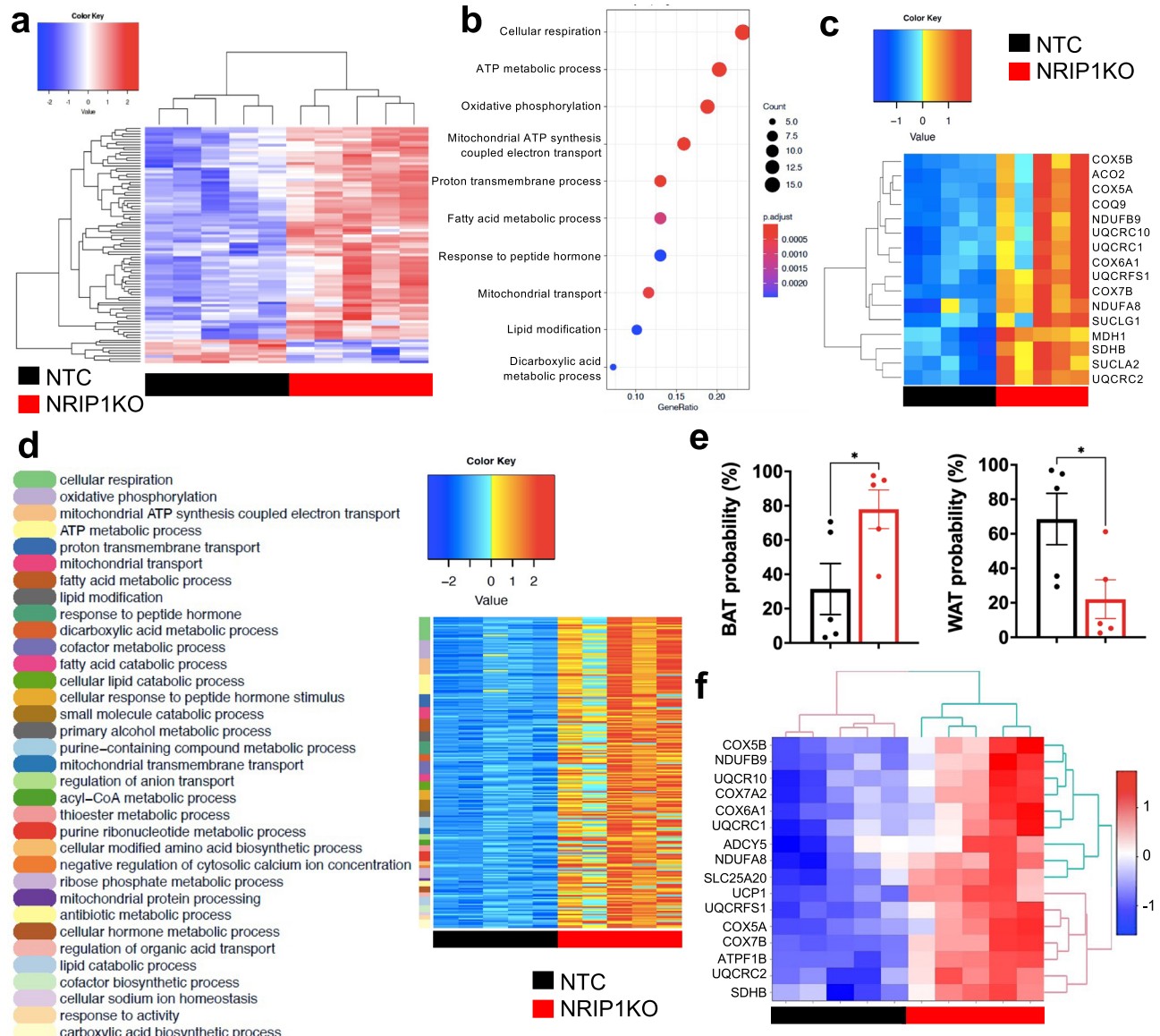

**Fig. 6 RNA sequencing on human NTC and NRIP1KO mature adipocytes on day 7 after differentiation. a** Heatmap of all differentially expressed genes between NTC and NRIP1KO based on normalized read counts. **b** Top 10 pathways associated with all upregulated genes detected. **c** Heatmap of genes associated with cellular respiration that were upregulated in NRIP1KO adipocytes. **d** Heatmap of pathway analysis of upregulated genes ranked by the *p* value. **e** Browning probability calculated using the ProFAT online tool. *$*P = 0.037$ by unpaired *T* test. **f** Heatmap of genes related to thermogenesis that were found highly expressed in the NRIP1KO samples compared to the NTC. Cut-offs were set at padj < 0.1 and fold change > 1.3. NTC ($n = 5$), NRIP1KO ($n = 5$). NRIP1KO adipocytes were transfected with SpyCas9 and *NRIP1* sgRNA-H5. Bars denote Mean and error bars denote Mean ± S.E.M.

sequencing (Supplementary Fig. 5). To further enhance the on-target effects and reduce off-target effects of the work described here, the use of high fidelity nucleases[44,45] can be explored and would further maximize cell viability and functionality. Altogether, the experiments reported here advance a powerful strategy for cell therapy in metabolic disease that can serve as a framework for testing in larger animals towards the longer-term goal of clinical trials.

## Methods

**Animals and diets**. All animal work was approved by the University of Massachusetts Medical School Institutional Animal Care Use Committee (IACUC protocol no.1600 to Michael P. Czech and no. 2007 to Silvia Corvera) with adherence to the laws of the United States and regulations of the Department of Agriculture. Mice were housed at 20–22 °C on a 12-hour light/12-hour dark cycle with ad libitum access to food and water. C57BL/6 J male mice were purchased from Jackson Laboratory for implant studies. C57BL/6 J (Jackson Laboratory) male mice were bred for primary preadipocyte cultures. Briefly, 10-week old male mice

arrived and were allowed to acclimate for a week prior to any procedures. Baseline glucose tolerance tests were performed three days before implanting edited cells. Mice were implanted with edited primary mouse adipocytes at 11 weeks of age by anesthetizing prior to the implantation procedure using an anesthesia vaporizer chamber with a continuous flow 500 cc/minute of $0_2$ with 3% (v/v) isoflurane for induction and 1.5% (v/v) for maintenance. After the cell injections, animals were allowed to wake up and were placed back in clean cages. Mice were maintained on a chow diet for the first 6 weeks, followed by a 60 kcal% fat diet (Research Diets, D12492i) for the remainder of the experiment from 6 to 16 weeks post implant. Glucose tolerance tests were performed after 16-hour overnight fasting with intraperitoneal injection of 1 g/kg D(+) glucose. Insulin tolerance tests were performed with 0.75 IU/kg after 6-hour daytime fasting. Male NOD.Cg-*Prkdc*scid *Il2rg*tm1Wjl/SzJ (denoted as NSG) mice were kindly donated by Taconic Biosciences, Inc. At 11 weeks of age NSG mice received implants with edited primary human adipocytes. Mice were maintained on a chow diet for the first 10 weeks, followed by placing them at thermoneutral environment with a 60 kcal% high fat diet (Research Diets, D12492i) for the remainder of the experiment from 10 to 15 weeks post implant. Housing under thermoneutrality was achieved by placing the NSG mice at 30 °C on a 12-hour light/12-hour dark cycle. Glucose tolerance tests with NSG mice were performed after a 16-hour fast with intraperitoneal injection 2 g/kg D(+)

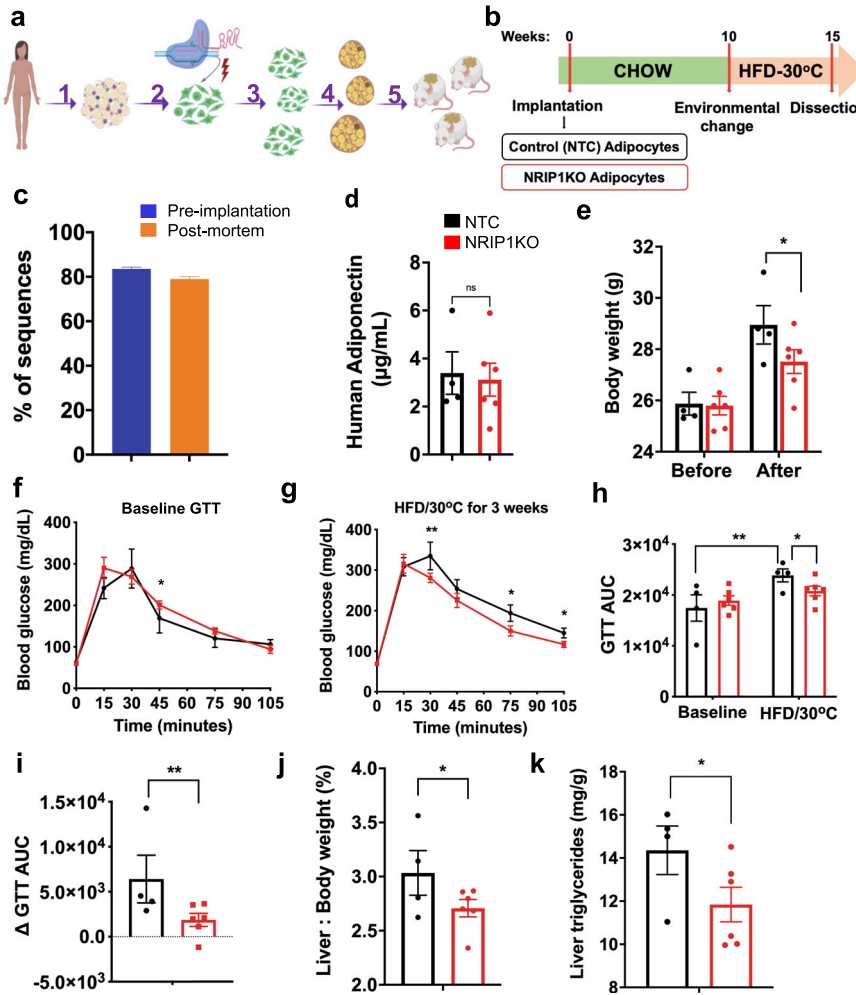

**Fig. 7 Implantation of NRIP1-targeted human adipocytes decreases body weight as well as liver triglyceride, and enhances glucose tolerance in recipient immunocompromised, HFD fed NSG mice.** Mice were implanted with either human adipocytes previously treated with NTC sgRNA/Cas9 RNPs or with sgRNA-H5/Cas9 RNPs (NRIP1KO). **a** Study description: (1) adipose tissue isolation from a human donor during panniculectomy, (2) harvesting of human primary preadipocytes after development Cas9/sgRNA RNPs were delivered into the human preadipocytes by electroporation followed by (3) expansion 1:6 of the genetically modified preadipocytes and (4) their differentiation into mature adipocytes; (5) Implantation of non-targeted control (NTC) sgRNA treated adipocytes versus the NRIP1 depleted adipocytes was performed in the dorsal area of NSG mice. **b** Schematic protocol of implantation of human NTC adipocytes or NRIP1KO adipocytes into NSG mice followed by HFD feeding. **c** Editing efficiency as evaluated with percentage of indels in the mature adipocytes transfected with NRIP1KO sgRNA-H5 before implantation (blue) and indel percentage in the genomic DNA isolated from the NRIP1 depleted implants 15 weeks following transplantation. **d** Human adiponectin levels detected in the plasma of NSG recipients 9 weeks following transplantation for the assessment of engraftment and functionality of the implants. $P = 0.812$ (ns). **e** Total body weight of NSG mouse recipients before transplantation (left) and after 3 weeks on HFD and thermoneutrality (right). **f** Baseline glucose tolerance test after 16 h. fasting before transplantation. $*p = 0.029$. **g** Glucose tolerance test after 3 weeks on HFD and thermoneutrality. $P$ values $**30$ min $= 0.006$; $*75$ min $= 0.023$; $*105$ min $= 0.017$. **h** Glucose tolerance areas under the curve (GTT AUC) before transplantations (left) and 3 weeks after HFD under thermoneutrality (right). $**p = 0.002$; $*p = 0.033$. **i** Matched difference of the GTT AUC before transplantations (left) and 3 weeks after HFD and thermoneutrality (right). $**p = 0.001$. **j** Liver over whole body weight percentage. $P = 0.012$. **k** Triglyceride measurements in pulverized liver extracts after dissection. $P = 0.036$. Black = NTC cell implant recipients ($n = 4$); red = NRIP1KO cell implant recipients ($n = 6$). Each $n$ represents number of biologically independent mice. Bars denote mean, error bars denote mean ± SEM. $*p < 0.05$, $**p < 0.01$ by unpaired $T$ test. NRIP1KO adipocytes were transfected with SpyCas9 and $NRIP1$ sgRNA-H5.

glucose. Whole blood was drawn and placed in EDTA-containing tubes from living mice with submandibular vein punctures under anesthesia as described above and in the end of the study with cardiac puncture. Plasma was extracted with centrifugation of whole blood for 15 min at 300 rcf at 4 °C.

**Primary mouse preadipocyte isolation, culture and differentiation to primary adipocytes.** 2 to 3 week old C57BL/6 J male or female mice were euthanized and inguinal fat tissue was harvested (including lymph node) and placed in HBSS buffer (Gibco #14025) plus 3% (w/v) bovine serum albumin (BSA) (American Bioanalytical). The protocol was carried out as described previously[34] with the following modifications: cells were incubated in 2 mg/mL collagenase (Sigma #C6885) in HBSS BSA 3% (w/v) for 20 min to digest the tissue. Cells were cultured to sub-confluence in complete media containing DMEM/F12 media (Gibco #11330), 1% (v/v) Penicillin/streptomycin, 10%

(v/v) Fetal bovine serum (FBS) (Atlanta Biologicals #S11550), 100 µg/mL Normocin (Invivogen #Ant-nr-1) at which time they were transfected with RNPs by electroporation and re-plated. Adipocyte differentiation was induced in the edited cells 24 h post confluence[28]. Briefly, adipogenic differentiation was initiated on day 0 with the addition of 5 µg/mL insulin, 1 µM dexamethasone, 0.5 mM 3-isobutyl-1-methylxanthine, 60 µM indomethacin and 1 µM rosiglitazone in the complete media. On day 2 (48 h later), the media was changed with complete media enriched with 5 µg/mL insulin and 1 µM rosiglitazone and on day 4 the media was changed with complete media enriched with 5 µg/mL insulin. By day 6 after differentiation the cells are considered fully differentiated and may continue to be cultured in complete media.

**Human subjects.** Abdominal subcutaneous adipose tissue was obtained from discarded tissue following panniculectomy. All subjects provided written consent to

the use of tissue and all procedures were approved by the University of Massachusetts Institutional Review Board (IRB#14734_13). The research described here is not considered human subjects research according to the criteria of the DHHS and FDA, as the cells used were from previously obtained samples of adipose tissue from de-identified human subjects. The researchers in this study had no direct interactions with these human subjects or their identities.

**Primary human preadipocyte isolation, culture and differentiation to primary adipocytes**. Human preadipocytes were from previously de-identified explants of human abdominal subcutaneous adipose tissue from panniculectomy samples. The explants had been embedded and cultured in Matrigel, and cell suspensions from capillary growth were obtained using dispase and plated on standard tissue culture plates. Growth and passaging of these cells was performed using EGM-2 MV. This method was demonstrated to provide a large expansion of adipocyte progenitors from the vascular-like projections growing out of small pieces of human adipose tissue[23]. Human adipocyte progenitors were transfected with RNPs by electroporation and plated at a density greater than 70% confluence to allow for further expansion. Cells were grown to confluence, then adipogenic differentiation media was added to induce adipogenesis[23,46]. To induce adipogenesis, we used a minimal adipogenic cocktail of DMEM enriched with 10% (v/v) FBS and penicillin/streptomycin, 0.5 mM 3-isobutyl-1-methylxanthine, 1 μM dexamethasone and 1 μg/mL insulin (MDI) for 72 h. The medium was then replaced with DMEM plus 10% FBS. Subsequently, 50% of the medium was replaced with fresh medium every other day[23,46]. On day 10 post differentiation, cells were harvested for implantation in NSG mice by treating with 0.5 mg/mL collagenase in 1x Trypsin to detach from culture plates.

**Transfection of primary preadipocytes (mouse and human) with RNPs**. For ribonucleoprotein (RNP) transfection, we used the Neon Transfection System 100 μL Kit (ThermoFisher, #MPK10096) and we prepared a mix consisting unless otherwise specified of sgRNA 4 μM (Synthego or IDT DNA) purified *Spy*Cas9 protein 3 μM PNA Bio, #CP02 or 3xNLS-SpCas9[47] (prepared by the Scot Wolfe laboratory) in Buffer R provided in the Neon Transfection System Kit. The cells were resuspended in Resuspension Buffer R for a final number of 0.5–6 × 10^6 cells per electroporation. For delivering the RNP complex into primary pre-adipocytes the electroporation parameters used were voltage 1350 V, width of pulse 30 ms; number of pulses 1 unless otherwise specified. The electroporated cells were placed in complete media immediately following transfection, expanded, grown to confluence and differentiated into mature adipocytes for downstream applications. We found these methods improved the viability of preadipocytes and adipocytes and their ability to differentiate over methods reported while our manuscript was in preparation[48]. The sequences for the sgRNAs used are reported in Table 1.

**Implantation of primary mouse and human adipocytes**. Primary mouse and human mature adipocytes on day 6 and 10 post differentiation, respectively, were washed twice with 1 x PBS. 0.5 mg/mL collagenase in 1 x trypsin was used to dissociate the cells from the plate. The detached cells are pelleted at 300 rcf for 10 min at room temperature. The cells were washed with 1 x PBS, pelleted, and the PBS was removed. Cell pellets were kept on ice for a brief time until implantation. Each mouse adipocyte pellet deriving from 1 × 150 mm fully confluent plate was mixed with matrigel (Corning® Matrigel® Growth Factor Reduced Basement Membrane Matrix, Phenol Red-free, LDEV-free # 356231) up to a total volume of 500 μL on ice and the cell and matrigel suspension (500 ± 20 μL) was drawn into a 1 mL syringe without the needle. The cell and Matrigel mixture was injected into the anesthetized mouse recipient with a 20 G needle by tenting the subcutaneous subscapular area, inserting the needle into the tented space and injecting at a slow but continuous rate to avoid cell rupture and solidification of the matrigel. The injection site was pinched gently for 1 min to allow the implant to solidify, followed by withdrawing the needle with a twisting motion. Each C57BL/6 J mouse recipient was injected with 2 × 150 mm plates of fully confluent murine adipocytes split into two bilateral injections in the subscapular area. Each NSG mouse recipient received 1 × 150 mm plate split into two 500 μL bilateral subcutaneous injections in the dorsal area as described above.

**DNA harvest from cells and tissue**. At two distinct time-points, 72 h following transfection and after primary adipocyte differentiation between day 6–10 post differentiation, genomic DNA was isolated from the transfected cells using DNA QuickExtract™ Buffer (Lucigen) in adherence to the manufacturer's instructions.

**Indel analysis by TIDE and ICE**. Genomic DNA was PCR amplified for downstream analysis using locus specific primers designed with MacVector 17.0 and purchased from IDT DNA and Genewiz, spanning the region 800 bp around the expected DSB. For the PCR, Kappa 2x Hot start HiFi mix was used and PCR products were purified using the QIAgen DNA purification kit following the manufacturer's instructions and submitted to Genewiz for Sanger Sequencing. Sanger sequencing trace data were analyzed with TIDE and ICE webtools (http://shinyapps.datacurators.nl/tide/, https://ice.synthego.com/#/) that decipher the composition of indels created at the sites of DSBs[49,50]. All primers used for the assessment of on-target and off-target editing are listed in Supplementary Table 1 in Supplementary Information file.

**RNA isolation**. Transfected cells were harvested for RNA between day 6–10, post-differentiation depending on the experiment by removing media and washing once with 1 x PBS, and adding Trizol reagent to lyse the cells. The protocol for RNA isolation was performed according to manufacturer's instruction with the following modifications: 1 μL of Glycol blue (Invitrogen #AM9516) was added to the isopropanol to precipitate the RNA and was either stored overnight at −20 °C or placed on dry ice for 2 h. The isolated RNA was resuspended in RNase free water, then treated with recombinant DNase I (DNA-free DNA removal kit, Ambion) according to the manufacturer's instructions. RNA concentrations were determined by Nanodrop 2000.

**RNA isolation of pulverized tissue/tissue piece**. Tissue was isolated from the mice and frozen in liquid N2. For RNA isolation, tissue was pulverized in liquid N2, or a piece approx. 100 mg in size was put in a 2 mL tube with screw cap and 1 mL of Trizol. Tissue was placed in the Qiagen TissueLyser and homogenized for 3 cycles of 3 min at 30 Hz. The Trizol and tissue lysate were placed in a new tube, and centrifuged for 10 min at 4 °C to separate any lipid from the homogenate. Once the homogenate is separated from the lipid, the remaining isolation is carried out according to manufacturer's instructions.

**RT-PCR**. 0.5−1 μg of RNA was used in 20 μL reaction with Bio-Rad iScript cDNA kit according to manufacturer's protocol to synthesize cDNA. cDNA was diluted by adding 80 μL of water to the reaction and 5 μL of cDNA template was used for RT-PCR with Bio-Rad Sybr Green Mix and gene specific primers for a final concentration of 0.3 μM primers. Expression of genes was determined by comparing gene expression levels of target gene compared to housekeeping gene 36B4 and RPL4 for murine and human samples, respectively. mRNA expression was analyzed with the ΔΔCT method. All primers used for RT-PCR are listed in Supplementary Table 2 in Supplementary Information file.

**Protein isolation**. Cells grown in culture dishes were washed once with 1 x PBS at room temperature, followed by adding boiling 2% SDS (w/v) with 1 x HALT protease inhibitors and scraping to lyse the cells. Tissue pieces were prepared for western blots by homogenizing a piece ~100 mg in Radioimmunoprecipitation Assay (RIPA) buffer with 1 x HALT protease inhibitors in the Qiagen TissueLyser and homogenized for 3 cycles of 3 min at 30 Hz. Tissue and cell lysates prepared with 2% SDS (w/v) buffer or RIPA buffer were sonicated at 60% amplitude with a probe sonicator tip for 30 s at room temperature. In Fig. 1b, mouse cells were lysed as described above at different time-points after transfection and for timepoint 0 h, after the electroporation the transfection mix consisting of cells and RNPs in Buffer R was centrifugated at 300 rcf. The cell pellet was lysed as described above while the supernatant (sup) was also collected for use as positive control (SpyCas9 3 μM) in the western blot. Protein concentration determination of the tissue and cell lysates was performed using a bicinchoninic acid kit (BCA Protein Assay Kit, Pierce). Cell lysates used in immunoprecipitation reaction were prepared in non-denaturing

---

| Table 1 Sequences of sgRNAs. | | | |
|---|---|---|---|
| **Mouse *Nrip1* sgRNA sequences (5′→ 3′)** | | **Human *NRIP1* sgRNA sequences (5′→ 3′)** | |
| sgRNA-M1 | CTTGTATTGAACATGACTCA | sgRNA-H1 | CTTCTATTGAACATGACTCA |
| sgRNA-M2 | ATTGTCTTAACTTACCTCGA | sgRNA-H2 | GCTTGGCTCTGATGTGCACC |
| sgRNA-M3 | GTCAGTACCCAGACGTACCA | sgRNA-H3 | ACACATACATATCAGGGGTC |
| sgRNA-M4 | ATAAGGTTTGGAGTCACGTC | sgRNA-H4 | ACATCAGGAAGATTCGTATC |
| sgRNA-M5 | CACTTTGTCCCACTGCGGGA | sgRNA-H5 | GTCATGTGCTGCAAGATTAC |
| sgRNA-M6 | ACAGGCTGTTGCCAGCATGG | sgRNA-H6 | TTTGCATGGTCCCTAAGAAA |
| sgRNA-M7 | GGAGTCGAAGAACATCTGCA | | |
| Human and Mouse Non targeting control sgRNA (NTC): GCACTACCAGAGCTAACTCA. | | | |

NP-40 buffer (20 mM Tris HCl pH 8.0, 137 mM NaCl, 1% (v/v) Nonident P-40 (NP-40), 2 mM EDTA) containing 1X HALT protease inhibitors by washing once with 1 x PBS, adding NP-40 buffer and scraping, followed by a 4 °C incubation for 30 min to 1 h with gentle agitation. Cell lysates were centrifuged at 4 °C for 10 min at 16,100 rcf and the infranatant was collected and used. Protein concentrations were determined on the lysates using Pierce BCA Kit. Protein samples were prepared for running on 7.5–12% SDS-PAGE mini gels at a final concentration of 1 mg/mL protein, 1 x Laemmli loading buffer (BioRad) with 2.5% (v/v) β-Mercaptoethanol, followed by placing in a heat block at 95 °C for 10 min.

**Oxygen consumption rate assay**. Oxygen consumption assay was performed using the Cayman Oxygen Consumption Rate Assay Kit (Cayman Chemical #600800) and measured using a Tecan Saffire II. Briefly, primary pre-adipocytes (male or female) were plated in 96-well flat bottom plates after transfection with NTC- or NRIP1 sgRNA-M6-RNPs. Cells were grown to confluence, followed by differentiation. On day 7 post differentiation, oxygen consumption assays were performed. All chemicals used in the assay were diluted in OCR media consisting of DMEM/F12 media (Gibco #11330), 10% (v/v) Fetal bovine serum (FBS) (Atlanta Biologicals #S11550), 2% (w/v) Fatty Acid Free BSA (Sigma #8806) and prepared as batch-working solutions to reduce pipetting error. The contribution of mitochondrial oxygen consumption was determined by subtracting values obtained in the presence of antimycin A (1:29 final dilution, Cayman Chemical kit). Oligomycin was used at 2.5 μM final concentration, FCCP at 5 μM and 10 μM L-(−)-Norepinephrine (+)-bitartrate salt (Sigma #N5785) was added where indicated. Each well was covered with 100 μL of mineral oil to seal the wells from ambient oxygen. Kinetic measurements were carried out with the fluorescent excitation/emission wavelengths of 380/650 nm, and readings taken every 2 min for 120 min. This assay was performed in both male and female primary adipocytes multiple times and the data shown here is representative of the assays performed. The final data shown is the average of 3 technical replicates at each time point with the corresponding average 3 technical replicates of the antimycin A sample subtracted to reflect only mitochondrial respiration values.

**Triglyceride assay**. For the liver triglyceride assay, we used the Triglyceride Colorimetric assay kit (Cayman Chemical, #10010303). The lysate was prepared by mixing 50 mg of pulverized whole liver with 1.5 mL of the NP-40 lysis buffer and homogenized in the Qiagen TissueLyser with 3 cycles 3 min at 30 Hz. The assay ran according to manufacturer instructions with a sample dilution of 1:5.

**Histology**. Approximately 0.5 cm$^2$ of the implant tissue and two 0.5 cm$^2$ liver pieces from two different lobes per recipient were randomly selected and fixed, followed by processing at the UMass Medical School Morphology Core. Photos of the tissues were taken with an LEICA DM 2500 LED inverted microscope at indicated magnification. Fiji/ImageJ was used to quantify lipid content in H&E images. 4 images per section, 2 sections per liver, were projected into a single montage. The montage was converted from RGB to 8 bit, contrast enhanced, thresholded and binarized. The processed montage was reconverted into individual images and lipid droplets quantified for each image using the particle analysis function (number, size, % of area covered).

**Western blotting and immunoprecipitation**. Protein lysates were run on 7.5% and 12% SDS-PAGE or Mini-Protean TGX stain-free pre-cast protein gels, followed by transferring the proteins to Nitrocellulose. Unless otherwise stated, a total of 20 μg of protein lysate was loaded per well. Nitrocellulose membranes were blocked using 5% (w/v) Non-fat milk in Tris buffered saline with 0.1% (v/v) Tween-20 (TBST) for 1 h at room temperature. Primary antibody incubations were carried out in 5% (w/v) BSA in TBST at the following antibody concentrations: UCP1-Abcam#10983, 1:700; Rip140-Millipore #MABS1917, 1:1000, Tubulin-Sigma #T5168, 1:4000; GAPDH-Cell Signaling #21185, 1:1000; SpyCas9-Cell Signaling #19526 S 1:5000, OXPHOS Abcam #110413, 1:1000. Blots and primary antibodies were incubated overnight with a roller mixer at 4 °C. Membranes were washed with TBST prior to secondary antibody incubations. HRP-conjugated secondary antibodies were diluted with 5% BSA (w/v) in TBST at 1:5,000–10,000 for 45 min at room temperature with constant shaking. Membranes were washed in TBST, followed by incubating with Perkin Elmer Western Lightning Enhance ECL. The Bio-Rad Chemi-Doc XRS was used to image the chemiluminescence and quantifications were performed using the system software, or Image J v.1.51. Immunoprecipitation was performed with NP-40/Halt protein lysates. Briefly, 250 μg of protein lysates were pre-cleared using 50:50 Protein-A Sepharose/NP-40 buffer/1 x HALT protease inhibitors for 2 h at 4 °C with end over end mixing. After 2 h, the lysate/Protein A Sepharose was centrifuged for 5 s to pellet the Sepharose, and the lysate was transferred to a new tube. 5 μg of Antibody (Rabbit Non-Immune IgG, Millipore #12-370, or Rabbit anti-Nrip1, Abcam #Ab42126) was added to the lysates and they were incubated overnight at 4 °C with end over end mixing. Antibody/antigen was pulled down by adding 50:50 Protein A Sepharose/NP-40 buffer/1 x HALT protease inhibitors for an additional 2 h at 4 °C with end over end mixing. Protein/Antibody/Protein A Sepharose complexes were washed by centrifuging briefly, removing the supernatant and washing the pellet with NP-40 buffer containing protease inhibitors. The captured proteins were eluted from

the Sepharose by adding 40 μL of 1 x Laemmli buffer containing 2.5% (v/v) β-Mercaptoethanol, vortexing the Sepharose mixture, followed by boiling at 95 °C for 10 min. All eluted proteins were run on the gel, transferred to nitrocellulose, and immunoblotted as described above.

**RNA-sequencing**. RNA was isolated on day 6 post differentiation and treated with DNase treatment as described, and 2 μg of total RNA was submitted to GENEWIZ for standard RNA sequencing (Illumina HiSeq). The data were acquired demultiplexed in the form of FASTQ files. Alignment and quantification of gene expression levels were performed using the DolphinNext RNA-seq pipeline (revision 3)[51]. Parameters for the pipeline were set to remove Illumina 3′ adapter sequences using trimmomatic software (v 0.39) with seed mismatches set to 2, a palindrome clip threshold of 30 and a simple clip threshold of 5[52]. The DolphinNext pipeline used RSEM (v1.3.1) to align RNA-Seq reads to a mouse reference transcript (using the RSEM reference STAR and and Bowtie genomes) to estimate gene expression levels[53]. DESeq2 software (v 1.28.1) was then used on these expression levels to find differentially expressed (DE) genes between two groups of samples. We set the parameters test = "LRT" (Likelihood Ratio Test), fitType = "parametric", betaPrior = FALSE, and reduced = ~1 (to compare to the control group). Alpha (padj) was set to 0.1 and a minFC = 1.3 was specified[54]. Once DE genes were found, enriched pathways were identified using the biomaRt (v 2.44.2) enrichGO routine. Parameters for this routine were set to orgDb = " org.Mm.eg.db", pAdjustMethod = "fdr", p valueCutoff = 0.05, ont = "BP", and minGSSize = 10[55]. The "universe" was set to all the genes found by RSEM. Similar pathways were combined using clusterProfiler's (v 3.16.1) simplify() function. The top 10 pathways were then displayed using the dotplot routine from the enrichplot package (version 1.8.1, loaded by clusterProfiler)[56]. Pathways where further culled or merged based on pathway names specified manually. Heatmaps of selected genes were generated using the normalized (for sample depth) values returned by the DESeq2 counts (ddsRES, normalized = TRUE) function. Prior to display values were standardized (each gene had its mean expression level subtracted and was then divided by the gene's standard deviation). The gplots package (v 3.1.0) was then used, via the heatmap.2 function, to display the heatmap. Custom software was written to combine lists of genes from many pathways to create the gene list used in the "stacked pathway heatmap", which was then displayed using heatmap.2. Principal component analysis was applied to the normalized data (DESeq2 getNormalizedMatrix() with method = "MRN"). The debrowser package's (v 1.16.3) run_pca() routine was used to calculate the principal components; it centers and scales the data prior to calculating the principal components[57–59].

**Human adiponectin**. Human adiponectin in the plasma of NSG mice was measured using a human-specific adiponectin ELISA from Invitrogen (KHP0041).

**Plasmid construction**. The pCS2-Dest plasmid with CMV promoter expressing SpyCas9 (Addgene # 69220), and sgRNA expressing plasmid (Addgene #52628) were a gift from Dr. Scot Wolfe lab. To clone NRIP1 targeting and non-targeting sgRNAs, oligo spacers with BfuAI overhangs (purchased from IDT) were annealed and cloned into the BfuAI-digested sgRNA plasmid. Lonza pmaxGFP LOT 2-00096 was used to test transfection of these plasmids in various concentrations to determine the efficient dosage range (0.5–1.5 μg) and the electroporation conditions (1350 V, 30 ms, 1 pulse) for the delivery and GFP expression was evaluated with EVOS FL fluorescent microscope (Thermo Fisher Scientific).

**Plasma cytokine and chemokine analysis**. Whole blood was collected from the mouse recipients with cardiac puncture at the end of the studies and placed in EDTA containing tubes. The plasma was collected after a 15 min centrifugation at 2000 rcf and 4 ºC. Multiplex analysis of plasma cytokines and chemokines was performed by the Luminex system. For positive controls, 4 C57BL6/J mice age and gender-matched to the recipients with body weights ranging from 34 to 40 g were injected with 1 μg LPS diluted in PBS and their plasma was collected as described above 2 h after injections. For the data analysis, the measurements found to be below the detectable cut-off were considered as 0 as a lowest value of 0.4 pg/mL was measured.

**GUIDE-seq**. In order to obtain results that correspond to our CRISPR delivery method and cell-type, we modified the GUIDE-seq transfection protocol[38] using our standard RNP concentration and electroporation method combined with the dsODN in various amounts (7.5, 10, 20, 30, 50, 80, 100 pmols) to achieve optimal on-target editing and dsODN integration. Three days after transfection, genomic DNA was isolated using the DNeasy Blood and Tissue kit (Qiagen) according to manufacturer's instructions. GUIDE-seq libraries were prepared using the custom oligos and adapters according to Table 2 with the sequences provided below. The barcoded libraries were sequenced on a MiniSeq platform in a paired end "147 | 8 | 16 | 147" run. Sequencer output was demultiplexed, trimmed and aligned according to a published workflow[60]. Positive- and negative-strand BAM files, along with a UMI reference, were processed as GUIDEseq[61] inputs with parameters allowing for an NGG PAM and 10 or fewer mismatches.
Index1: ATCACCGACTGCCCATAGAGAGGACTCCAGTCAC
Read2: GTGACTGGAGTCCTCTCTATGGGCAGTCGGTGAT

**Table 2 GUIDE-Seq library set up.**

| Sample info | Fastq file name | Strand | Index-2 | Index-1 | Index-2 | Index-1 |
|---|---|---|---|---|---|---|
| Mouse dsODN | | Positive | A01 | P701 | TAGATCGC | TCGCCTTA |
| Mouse dsODN + RNP-M6 | | | A02 | + | CTCTCTAT | + |
| Human dsODN | ET3B | | A03 | P702 | TATCCTCT | CTAGTACG |
| Human dsODN 10 pmol + RNP-H5 | ET4B | | A04 | + | AGAGTAGA | + |
| Human dsODN 7.5 pmol + RNP-H5 | ET6A | | A02* | P703 | CTCTCTAT | TTCTGCCT |
| | | | | + | | + |
| | | | | P704 | | GCTCAGGA |
| | | | | | | |
| Mouse dsODN | | Negative | A01 | P705 | TAGATCGC | AGGAGTCC |
| Mouse dsODN + RNP-M6 | | | A02 | + | CTCTCTAT | + |
| Human dsODN | ET3B | | A03 | P706 | TATCCTCT | CATGCCTA |
| Human dsODN 10 pmol + RNP-H5 | ET4B | | A04 | + | AGAGTAGA | + |
| Human dsODN 7.5 pmol + RNP-H5 | ET6A | | A02* | P707 | CTCTCTAT | GTAGAGAG |
| | | | | + | | + |
| | | | | P708 | | CCTCTCTG |

*Separate run

P7 Adapters Sequence (5′ → 3′) (Index 2):

P701: CAAGCAGAAGACGGCATACGAGAT**TCGCCTTA**GTGACTGGAGTCCTCTCTATGGGCAGTCGGTGA

P702: CAAGCAGAAGACGGCATACGAGAT**CTAGTACG**GTGACTGGAGTCCTCTCTATGGGCAGTCGGTGA

P703: CAAGCAGAAGACGGCATACGAGAT**TTCTGCCT**GTGACTGGAGTCCTCTCTATGGGCAGTCGGTGA

P704: CAAGCAGAAGACGGCATACGAGAT**GCTCAGGA**GTGACTGGAGTCCTCTCTATGGGCAGTCGGTGA

P705: CAAGCAGAAGACGGCATACGAGAT**AGGAGTCC**GTGACTGGAGTCCTCTCTATGGGCAGTCGGTGA

P706: CAAGCAGAAGACGGCATACGAGAT**CATGCCTA**GTGACTGGAGTCCTCTCTATGGGCAGTCGGTGA

P707: CAAGCAGAAGACGGCATACGAGAT**GTAGAGAG**GTGACTGGAGTCCTCTCTATGGGCAGTCGGTGA

P708: CAAGCAGAAGACGGCATACGAGAT**CCTCTCTG**GTGACTGGAGTCCTCTCTATGGGCAGTCGGTGA

Adapter Sequences (Index 1):

A01: AATGATACGGCGACCACCGAGATCTACACT**AGATCGC**NNWNNWNNACACTCTTTCCCTACACGACGCTCTTCCGATC*T

A02: AATGATACGGCGACCACCGAGATCTACACC**TCTCTAT**NNWNNWNNACACTCTTTCCCTACACGACGCTCTTCCGATC*T

A03: AATGATACGGCGACCACCGAGATCTACACT**ATCCTCT**NNWNNWNNACACTCTTTCCCTACACGACGCTCTTCCGATC*T

A04: AATGATACGGCGACCACCGAGATCTACACA**GAGTAGA**NNWNNWNNACACTCTTTCCCTACACGACGCTCTTCCGATC*T

**Amplicon sequencing analysis of on- and off-target editing.** Genomic DNA was isolated from both human and mouse NTC or NRIP1KO adipocytes after day 6 post differentiation $n$ [NTC] = 3; $n$ [NRIP1KO] = 3 per amplicon. Illumina amplicon sequencing library was prepared using a two-step PCR protocol. During PCR1, regions of interest (around 290 bp) were amplified as follows: 98 °C for 2 min, 24 cycles of 98 °C for 15 sec −64 °C for 20 sec −72 °C for 15 sec, and 72 °C for 5 min, in a reaction mix of 50 ng genomic DNA, 10 µM forward and reverse primers that contain Illumina adapter sequences 1 µL each, 12.5 µL NEBNext UltraII Q5 Master Mix, and water to bring the total volume to 25 µL. PCR1 primer sequences are in Supplementary Table 1 in the Supplementary information file. Then, PCR2 was performed as follows: 98 °C for 2 min, 10 cycles of 98 °C for 15 sec −64 °C for 20 sec −72 °C for 15 sec, and 72 °C for 5 min, in a reaction mix of 5 µL of total PCR1 product, 10 µM forward and reverse primers that contain unique barcode sequences 2 µL each, 25 µL NEBNext UltraII Q5 Master Mix, and water to bring the total volume to 50 µL. Index sequences used in PCR2 are in Table 3. PCR2 products were first purified (Zymo PCR purification kit), visualized using 1.5% agarose gel electrophoresis, and pooled together for similar amounts based on the band intensities. Pooled PCR2 products ran on 2% agarose gel electrophoresis and cut for desired bands (around 400 bp) for gel extraction (Zymoclean DNA Gel recovery kit). Concentration of the final purified library was determined using Qubit (High Sensitivity DNA assay). The integrity of the library was confirmed by Agilent Tapestation using Agilent High Sensitivity D1000 ScreenTape kit. The library was sequenced on an Illumina Miniseq platform according to the manufacturer's instructions using Miniseq Mid Output Kit (300-cycles) in a paired end "151 | 6 | 8 | 151" run. CRISPResso2 was used to align the reads and quantify the editing[62].

**Statistics and reproducibility.** Experiments with primary mouse adipocytes in vitro were performed multiple times with sgRNA-M6 by independent investigators. For the experiments in Figure panels 1c, d, e, h, i we performed biologically independent replicates between 2 and 8 times. In Fig. 1g and 1i, we performed biological replicates between 2 and 4 times depending on the guide used. In Fig. 2c we performed the titration of cas9 and sgRNA one time. In Fig. 2d the oxidative phosphorylation western blot was performed one time with all guides represented in a single experiment and 4 times with guide NTC and sgRNA-M6. Figure panels 2e–j oxygen consumption analysis was performed in male primary adipocytes 4 times and 2 times in female primary adipocytes. Regarding Fig. 3 and Supplementary Fig. 4, four independent cohorts were studied sequentially in time following the same protocol as described above. Each of these cohorts included both conditions (NTC, NRIP1KO murine implants) with the following number of mice: in cohort 1 included n (NTC) = 2 and $n$ (NRIP1KO) = 2, in cohort 2 $n$ (NTC) = 2 and $n$ (NRIP1KO) = 1, in cohort 3 n (NTC) = 3 and $n$ (NRIP1KO) = 5, in cohort 4 $n$ (NTC) = 6 and $n$ (NRIP1KO) = 6 where $n$ represents a population consisting of separate mice. The phenotype described in this study was consistent among all cohorts. All mice were followed up with the same protocol during the study including body weight measurements, glucose tolerance tests, while in post mortem the animals were split into different analysis. Macroscopic and histology images are representative. For steatosis quantification in H&E sections, multiple images of randomly selected sections were used at three different levels of the liver preparations. In Figs. 4 and 6, RNA samples for RNA sequencing were submitted from biologically independent experiments. Sample images shown in Fig. 5d of NRIP1KO in human primary adipocytes has been performed up to 4 times. For the experiments in Figure panels 5b, c, d, e, g we performed biologically independent replicates between 2 and 8 times. In vitro experiment Fig. 5f was performed two times. Figure 5h experiment was performed with 2 biological replicates and one of the sets of biological replicates was run in the western blot one time. In vivo data presented in Fig. 7 derive from one cohort of NGS mice which included both conditions with sample size (NTC) = 4 and $n$ (NRIP1KO) = 6 where n represents a population consisting of separate mice. All comparisons between two groups were performed with student unpaired two-tailed $T$ Test with the following demonstration of p values in the panels: $*p < 0.05$, $**p < 0.01$, $***p < 0.001$. In data that normal distribution cannot be assumed or proven, normalization to equal standard distributions preceded the statistical analysis. All $p$ values, $t$ values and degrees of freedom are reported in the raw data file. All comparisons between more than two groups (Fig. 4, Supplementary Fig. 4k) were performed with one-way ANOVA and multiple comparison's in those were performed with Dunnett's multiple comparisons test. P value, F value are provided in the raw data file. Statistics used in the oxygen consumption analysis (Fig. 2e, f, h, i) were One-way ANOVA with Sidak's multiple comparison test and two—way ANOVA for the 40 min timepoint summary (Fig. 2g, h) with Sidak's multiple comparison test.

**Databases, software and online tools.** For the mapping of exons on the Nrip1 gene we used IGV_2.5.3. For the Design of sgRNAs we used a combination of the

**Table 3 Amplicon NGS sequencing Indexes.**

3′ Index sequences

| | |
|---|---|
| i301 | CAAGCAGAAGACGGCATACGAGATCGTGATGTGACTGGAGTTCAGACGTGTGCTCTTCCG |
| i302 | CAAGCAGAAGACGGCATACGAGATACATCGGTGACTGGAGTTCAGACGTGTGCTCTTCCG |
| i303 | CAAGCAGAAGACGGCATACGAGATGCCTAAGTGACTGGAGTTCAGACGTGTGCTCTTCCG |
| i304 | CAAGCAGAAGACGGCATACGAGATTGGTCAGTGACTGGAGTTCAGACGTGTGCTCTTCCG |
| i305 | CAAGCAGAAGACGGCATACGAGATCACTGTGTGACTGGAGTTCAGACGTGTGCTCTTCCG |
| i306 | CAAGCAGAAGACGGCATACGAGATATTGGCGTGACTGGAGTTCAGACGTGTGCTCTTCCG |
| i307 | CAAGCAGAAGACGGCATACGAGATGATCTGGTGACTGGAGTTCAGACGTGTGCTCTTCCG |
| i308 | CAAGCAGAAGACGGCATACGAGATTCAAGTGTGACTGGAGTTCAGACGTGTGCTCTTCCG |
| i309 | CAAGCAGAAGACGGCATACGAGATCTGATCGTGACTGGAGTTCAGACGTGTGCTCTTCCG |
| i310 | CAAGCAGAAGACGGCATACGAGATAAGCTAGTGACTGGAGTTCAGACGTGTGCTCTTCCG |
| i311 | CAAGCAGAAGACGGCATACGAGATGTAGCCGTGACTGGAGTTCAGACGTGTGCTCTTCCG |
| i312 | CAAGCAGAAGACGGCATACGAGATTACAAGGTGACTGGAGTTCAGACGTGTGCTCTTCCG |
| i313 | CAAGCAGAAGACGGCATACGAGATTTGACTGTGACTGGAGTTCAGACGTGTGCTCTTCCG |
| i314 | CAAGCAGAAGACGGCATACGAGATGGAACTGTGACTGGAGTTCAGACGTGTGCTCTTCCG |
| i315 | CAAGCAGAAGACGGCATACGAGATTGACATGTGACTGGAGTTCAGACGTGTGCTCTTCCG |
| i316 | CAAGCAGAAGACGGCATACGAGATGGACGGGTGACTGGAGTTCAGACGTGTGCTCTTCCG |
| i318 | CAAGCAGAAGACGGCATACGAGATGCGGACGTGACTGGAGTTCAGACGTGTGCTCTTCCG |
| i319 | CAAGCAGAAGACGGCATACGAGATTTTCACGTGACTGGAGTTCAGACGTGTGCTCTTCCG |

5′ Index sequences

| | |
|---|---|
| i501 | AATGATACGGCGACCACCGAGATCTACACTATAGCCTACACTCTTTCCCTACACGACGCTCTTCCGATCT |
| i502 | AATGATACGGCGACCACCGAGATCTACACATAGAGGCACACTCTTTCCCTACACGACGCTCTTCCGATCT |
| i503 | AATGATACGGCGACCACCGAGATCTACACCCTATCCTACACTCTTTCCCTACACGACGCTCTTCCGATCT |
| i504 | AATGATACGGCGACCACCGAGATCTACACGGCTCTGAACACTCTTTCCCTACACGACGCTCTTCCGATCT |
| i505 | AATGATACGGCGACCACCGAGATCTACACAGGCGAAGACACTCTTTCCCTACACGACGCTCTTCCGATCT |
| i506 | AATGATACGGCGACCACCGAGATCTACACTAATCTTAACACTCTTTCCCTACACGACGCTCTTCCGATCT |
| i507 | AATGATACGGCGACCACCGAGATCTACACCAGGACGTACACTCTTTCCCTACACGACGCTCTTCCGATCT |
| i508 | AATGATACGGCGACCACCGAGATCTACACGTACTGACACACTCTTTCCCTACACGACGCTCTTCCGATCT |
| i509 | AATGATACGGCGACCACCGAGATCTACACGACGACCTACACTCTTTCCCTACACGACGCTCTTCCGATCT |
| i510 | AATGATACGGCGACCACCGAGATCTACACTAATCGGCACACTCTTTCCCTACACGACGCTCTTCCGATCT |

Broad Institute sgRNA designer, CHOPCHOP and the online sgRNA checkers by Synthego and IDT. For the design of genomic DNA primers we used MacVector 17.0. For the alignment of the Sanger Sequencing traces and the human and mouse coding region we used SnapGene Viewer 5.1.6 and NCBI nucleotide blast. For the design of RT-PCR primers we used Primer Bank (https://pga.mgh.harvard.edu/primerbank/). For the browning probability potential, the unnormalized mapped read counts were applied in the ProFAT online tool[37]. Off-target diagrams were generated from GUIDEseq output using a freely available visualization tool (https://mismatch.netlify.app). For the data graphing, we used Prism GraphPad 9 unless otherwise specified.

**Reporting summary**. Further information on research design is available in the Nature Research Reporting Summary linked to this article.

## Data availability

All sequencing data that support the findings of this study have been deposited in the NIH Sequence Read Archive via BioProject PRJNA745932. Other source data are provided with this paper as a Source Data file[63]. Source data are provided with this paper.

## Code availability

All analysis was performed using publicly available programs and the parameters indicated in the "Methods" section.

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

## Acknowledgements

We wish to thank members of the Czech and Corvera laboratories for helpful discussions and Kerri Miller for excellent assistance in preparing the paper. We thank the University of Massachusetts Morphology Core Facility for assistance in the histological preparations, stains and analysis. This work was supported by the Assistant Secretary of Defense for Health Affairs endorsed by the Department of Defense, through the Peer Reviewed Medical Research Program under Award No. W81XWH-18-1-0397 and W81XWH-18-1-0398 (to S.C. and M.P.C.). Opinions, interpretations, conclusions and recommendations are those of the authors and are not necessarily endorsed by the Department of Defense. This work was also supported by National Institutes of Health grants DK030898 (to M.P.C.), GM115911 (to E.J.S. and S.A.W.), TR002668 (to E.J.S. and S.A.W.), HL147482 (to K.L.), the UMASS Mouse Metabolic Phenotyping Center at UMass Medical School (NIH grant 5U2C-DK093000 to J.K.K.). We also gratefully acknowledge generous funding through the Isadore and Fannie Foxman Chair in Medical Science (to M.P.C.), the Endowed Professorship in Diabetes Research Chair (to S.C.) and postdoctoral fellowship support to Felipe Henriques by the American Diabetes Foundation (Grant 1-19-PMF-035). Multiple figures in this paper (Fig. 1a, Fig. 5a, Fig. 7a, Supplementary Fig. 2b bottom) were created with Biorender.com.

## Author contributions

E.T., S.M.N., S.C. and M.P.C. designed the study, interpreted the data and wrote the paper. E.T. and S.N. performed most of the experiments, analyzed the data and performed the mouse adipocyte implantation studies. E.T., T.D.S. and J.SR. performed the human adipocyte implantation studies. S.M.N. performed the oxygen consumption assays. T.D.S., J.SR. and A.D. established the human adipose explant-derived cell lines. Y.S. contributed to the initial strategy of the work. M.K. established the colonies for mouse cell donors and performed the blood collections of the recipient mice. A.G. and F.H. performed experiments and guided methods. L.M.L. and E.T. analyzed the RNA-seq data. N.A., R.I. and E.J.S. guided methods to gene-editing optimization, GUIDE-seq and Amplicon NGS library preparations, cloning experiments and interpretation of data. E.T. prepared the GUIDE-seq and Amplicon NGS libraries, T.C.R. and L.M.L. analyzed the data GUIDE-seq data and T.C.R. analyzed the Amplicon NGS data. K.L., S.M. and S.A.W. purified SpyCas9 protein and developed the plasmids. R.H.F., L.T., X.H. and J.K.K. performed the measurement of cytokines/chemokines in submitted plasma samples. All authors reviewed and were invited to edit the paper.

## Competing interests

The authors declare the following competing Interests: M.P.C. and A.J.G. are inventors of granted US Patent #8,519,118, "RIP140 regulation of glucose transport", and of granted US Patent #8,093,223, "RIP140 regulation of diabetes", related to data in this paper on Nrip1/RIP140-depleted mouse and human adipocytes. SC is inventor on granted US Patent #10,093,902, "Human adipose tissue white and brown-on-white progenitors for

reconstructive and metabolic therapies", related to data in this paper on human adipose progenitors and human adipocytes. M.P.C., S.C., E.T., and S.M.N. are inventors of pending US Patent application #63/089,955, "Targeting Nrip1 to Alleviate Metabolic Disease", related to CRISPR-based depletion of Nrip1 in mouse and human adipocytes in this paper. The University of Massachusetts is the grantee or potential grantee of all of the above. MPC and SC declare that they are bound by confidentiality agreements that prevent them from disclosing a potentially competing interest in this work. E.J.S. is a co-founder and scientific advisor of Intellia Therapeutics. The remaining authors declare no competing interests.
