## [Peer Review File · Nature Communications]

REVIEWER COMMENTS

Reviewer #1 (Remarks to the Author):

Increasing the thermogenic adipocytes mass via cell transplantation in obese animal models has been proved to have beneficial effects on glucose tolerance and insulin sensitivity. In this field, an abundant source of human thermogenic adipocytes for cell-based therapy is still a challenge.

NRIP1 is a nuclear co repressor involved in several energy metabolic pathways, including in the energy expenditure balance. It has been previously published, by the authors of the submitted manuscript, being of white adipocytes deleted for Nrip 1 gene using the CRISPR-Cas9 technology (Yuefei Shen et al. JBC 2018). More recently, Yu-Hua Tseng's group has reported the use of a variant of CRISPR-Cas9 to create brown-like adipocytes (HUMBLE cells) by activation of endogenous UCP1 expression in immortalized human white adipocytes. Then, the therapeutic potential of HUMBLE adipocytes for metabolic disorders has been demonstrated in immunodeficient nude mice (Chih-Hao Wang et al. Sci.Transl. Med. 2020).

The major claim of the submitted manuscript is to overcome problems such as immune response upon implantation of cells created by CRISP-based methods using expression vectors or lentiviral transduction and long term maintenance of gene modifications. For that purpose, the authors developed a method to deliver ribonucleoprotein complexes of Cas9 and sgRNA by electroporation for ablation of Nrip 1 gene in murin and primary human adipose progenitors. Deletion efficiency is very high, and deficient adipocytes enhanced a glucose tolerance and improved metabolic parameters in implanted mice.

Major points:

1) Novelty of submitted data, compared to the two publications mentioned above, is not clearly exposed. The use of primary human adipose progenitors, and not of immortalized human cell lines could be one.

2) Authors claim that advantages of the method developed is to overcome immunogenicity that could be caused by CRISPR reagents, but this potential advantage is not directly addressed in the manuscript.

3) Visualisation of mouse and human adipocytes 16 weeks after implantation should partially address the concern of point 2)

4) Fig.1 g,h: Expression of UCP1 in Nrip1 targeted cells is not sufficient to claim induction of "thermogenic responses". As indicated by the authors, targeting NRP1 has the potential advantage

of upregulate expression of many genes that have favourable metabolic effects, compared to upregulation of UCP1 only as described by Chih-Hao Wang et al. Functional in vitro tests for energy expenditure cells and expression of thermogenic genes (more than those shown in Extended data Fig.4, such as Dio2, ADRB3, Glut1, PGC1a,..) have to be performed.

5) Targeted human primary adipose progenitors is a strong point of the manuscript. Targeted cells should be more characterized.

Minor point:

Figure 2f: Time of implant harvest is not indicated

Reviewer #2 (Remarks to the Author):

In this manuscript, Dr. Corvera/Dr. Czech and colleagues develop a highly efficient method to disrupt the nuclear co-repressor NRIP1 gene using CRISPR in mouse or human adipocytes. This is a follow-up of previous work (Shen et al, ref 34 in the manuscript). Disruption of the Nrip1 gene in white adipocytes markedly increases UCP1 gene expression. Here, the authors demonstrate the usefulness of delivering ribonucleoprotein complexes of SpyCas9 protein and single-guide RNA by electroporation transfection. Disruption of the Nrip1 gene at exon 4 is highly efficient, although the degree of NRIP1 protein loss is variable but highly correlated with upregulation of UCP1 expression. Moreover, using in vivo adipocyte transplant approaches previously developed by the authors (Min et al, ref 33), transfected primary mouse or human adipocytes were implanted in diet-induced obese mouse models. The impact of 'increased browning' (implanted NRIP1 deficient adipocytes highly expressing UCP1) is shown by improved glucose tolerance and decreased liver steatosis. Thus, the present study reveals the suitability of gene-engineered beige adipocyte cell therapy approaches for metabolic diseases. In fact, this is an emerging issue here and now (DOI: 10.1038/nrendo.2010.20): expanding the amount of active BAT and/or browning via transplantation holds promise as a therapeutic strategy for morbid obesity and diabetes. Several strategies have been pursued (BAT engraftment, brown adipose progenitor cell transplant), but very few genetic-modification experimental approaches have been reported to date (see refs 37, 45). Overall, the data presented in the manuscript is highly relevant and results are discussed appropriately in relation to the currently available literature. The reviewer has a few suggestions that could improve data presentation if implemented.

-Introduction. Indicate that NRIP1 is also denoted as RIP140.

-Fig 2. Characterization of implanted adipose tissue depots at study termination (16 weeks after implantation) should include other thermogenic/beige specific marker genes to see whether they

also retained elevated expression as shown for UCP1. In the legend, the number of biological replicates is confusing (c-e).

-Fig 4. As above, the characterization of implanted adipose tissue is required at study termination (13 weeks after implantation of genetically-engineered human adipocytes).

-NSG mouse model. Mice were on HFD for 5 weeks but also were placed at thermoneutrality. Why? To increase HFD-induced obesity and/or susceptibility to the effects of knock-down NRIP1 on UCP1 induction?

-Authors should discuss why despite similar high efficiency of Nrip1 gene disruption at 7 loci by SpyCas9/sgRNA RNPs, that results in variable degrees of NRIP1 protein loss in murine primary adipocytes. Does it also occur in the screen of sgRNAs targeting different loci of NRIP1 in human adipocytes?

-Methods. The protocol of primary mouse preadipocyte differentiation in culture should be detailed. At what day post-differentiation are the cells considered fully differentiated adipocytes? In the legend to Fig 1, analysis in mature primary adipocytes on day 6 post differentiation means day 6 after induction of differentiation or 6 days after cells reached differentiation? Regarding electroporation of adipocytes, has the percentage of cell death due to the electroporation procedure been measured?

-References. Several references lack 'et al'

Reviewer #3 (Remarks to the Author):

E Tsagkaraki et al. provide a powerful strategy for cell therapy in metabolic disease, consisting to deliver complexes of SpyCas9 protein and sgRNA ex-vivo targeting NRIP1 gene. The method is attractive but its application to human pathology requires further investigations and authors do not clearly explain how it could be envisageable.

Specific questions follow:

1- The reasons why the authors choose to target NRIP1 are not very clear and need to be clarify in introduction.

2- Conclusions appear sometimes not completely appropriate (Figure 2d) with results, in particular concerning the effects on glucose tolerance after NRIP1KO adipocytes implantation.

3- What are the long-term consequences of high UCP1 expression in adipocytes? Are adipocytes still able to respond to acute stress?

4- Authors should identify if side effects on the mouse aging (more than 45 old-weeks) could be observed

5- Authors only used male mice. Due to the effects of NRIP1 on estrogen receptors, authors should perform same experiments with adipocytes from female mice ?

6- Why authors did not perform adipocytes implantation directly in WAT or BAT tissues ?

7- Concerning human subjects, what are the clinical characteristics :age, sexe... ?

Reviewer #4 (Remarks to the Author):

This paper focuses on methods that deliver CRISPR systems ex vivo to disrupt the thermogenesis suppressor gene NRIP1 in human and mouse adipocytes. The focus is to improve metabolic homeostasis through the modification of human adipocytes without exposure of the recipient to immunogenic Cas9 or delivery vectors. The paper is well written, and the experimental design and flow is OK.

The genome editing experiments are performed well and the SANGER sequencing followed by analysis of the mRNA and protein levels of NRIP1 is appropriate and conclusive. I would be interested in seeing additional off-target analysis using guide-seq or deep sequencing. Using RNAi as control to downregulate NRIP1 would also be a valuable control.

My most important concern has to do with the novelty of this approach and manuscript. More specifically, and I am including representative papers (not a complete list), recent results show direct tissue-grafting approach to increasing endogenous brown fat (PMID: 29785004) or through editing of lncRNAs (PMCID: PMC5899897) or through CRISPR-based editing of uncoupling protein 1 expression (PMID: 32848096). Moreover, the delivery of CRISPR to fat tissue is not novel (PMID: 31467027) nor is the observation that NRIP1 is important (shown by the same group, PMID: 30190322).

While the authors attribute the relevance of upregulating NRIP1 to the resulting expression of many genes that have "favorable" metabolic effects, this is not clearly shown (with data) in the manuscript. Furthermore, I would argue that the comparison with UCP1 should be performed and shown in the manuscript. To conclude, considering the recent literature results, I don't believe the paper presents a sufficient advance to warrant publication in nature communications.

Response to Reviewers' Comments:

General Response to the Reviewers: We appreciate the general enthusiasm in most reviews as well as the comments on how to improve the study and manuscript. We therefore have attempted to address all the concerns and make the final paper as strong as possible within a reasonable timeframe. We have spent the last 6 months since submission performing additional key experiments in order to achieve maximal support for the conclusions and to expand the impact of the paper as suggested by the reviewers. In doing so, we have generated extensive additional data that addresses virtually all the comments of the reviewers. These new extensive results are included in 3 additional multi-panel Figures, in new panels within old Figures and in the new or revised Extended Figures 3 and 4 in the revised version. We have also revised the text in accordance with the reviewers' comments.

We are thankful to the reviewers for the very helpful comments which have facilitated our efforts to greatly improve the study for publication in Nature Communications. Our responses to the reviewers' comments and descriptions of the new data included are detailed below:

Reviewer 1:

Increasing the thermogenic adipocytes mass via cell transplantation in obese animal models has been proved to have beneficial effects on glucose tolerance and insulin sensitivity. In this field, an abundant source of human thermogenic adipocytes for cell-based therapy is still a challenge.

Response: We thank the reviewer for pointing out that a major feature of our study—expansion of human thermogenic adipocytes for cell-based therapy—is a challenge in the field. We feel our manuscript makes a major advance in this regard—that favorable CRISPR-based modifications can be made in expanded primary human cells.

The major claim of the submitted manuscript is to overcome problems such as immune response upon implantation of cells created by CRISPR-based methods using expression vectors or lentiviral transduction and long term maintenance of gene modifications. For that purpose, the authors developed a method to deliver ribonucleoprotein complexes of Cas9 and sgRNA by electroporation for ablation of Nrip 1 gene in murin and primary human adipose progenitors. Deletion efficiency is very high, and deficient adipocytes enhanced a glucose tolerance and improved metabolic parameters in implanted mice.

Response: Again, we thank the reviewer for these comments indicating that our data showing our method for delivery of Cas9/sgRNA yields very high efficiency and Nrip1 deficient human adipocytes improve metabolic parameters in mice.

1) Novelty of submitted data, compared to the two publications mentioned above, is not clearly exposed. The use of primary human adipose progenitors, and not of immortalized human cell lines could be one.

Response: This is a really important point that is critical to the work, and we thank the reviewer for noting that we have understated the novelty of our approach and results. While the papers cited are of keen interest, our work is instead devoted to advancing conditions of procuring and genetically modifying human adipocytes in the context of a therapeutic strategy. Previous work is not in this mode, as primary human cells from expanded progenitors were not used and methods of transfection were performed with vectors with continuous expression of immunogenic Cas9. As we now emphasize in the revised Abstract, Introduction and Discussion sections of the resubmitted manuscript, we report on methods developed and applied that can be much more easily advanced to a therapeutic stage. We now explicitly state the 5 criteria we have used in our studies that emphasize the novelty of our findings. Thus, we now state in the Discussion section:

“A major goal of the present studies was to advance the application of CRISPR technology to metabolic disease in the context of a potential therapeutic strategy. The starting point of our experimental plan was the success of many laboratories in demonstrating the efficacy of implanting mouse or human brown/beige adipocytes into glucose intolerant mice to alleviate diabetes. Five key criteria were incorporated into our approach to advance towards therapy: 1. Generation of large numbers of adipocyte progenitors from small samples of human adipose tissue, 2. Identification of a strong suppressor of adipocyte beiging for targeting by CRISPR to optimize the therapeutic benefit of adipocyte implantation, 3. Stealth administration of SpyCas9/sgRNA to cells that would not expose recipient mice to immunogenic reagents, 4. Minimizing off target effects, and 5. High efficiency gene disruption in which most all cells *ex vivo* are affected in a single step without a cell selection step. Taken together, the data presented here show that our methods using CRISPR-based RNPs targeting *Nrip1* to a large extent satisfy the above criteria. These methods can indeed enhance browning of mouse or human adipocytes *ex vivo* at high efficiency without the use of expression vectors to improve metabolism in obese mice.

2) Authors claim that advantages of the method developed is to overcome immunogenicity that could be caused by CRISPR reagents, but this potential advantage is not directly addressed in the manuscript.

3) Visualisation of mouse and human adipocytes 16 weeks after implantation should partially address the concern of point 2)

Response: Our objective on this important point of the reviewer’s was to devise a transfection method for Cas9/sgRNA RNPs that would have 2 key features—high efficiency in the entire plate of cells without the need for selection of transfected cells and loss of the Cas9 protein over a few days prior to differentiation into adipocytes and implantation. We have demonstrated success in these two objectives, and to the reviewer’s point have now moved the data on the loss of Cas9 protein to Figure 1 (panel b) from the original supplemental Figure to emphasize this key point. The absence of detectable Cas9 in the implanted cells (and therefore not available to mount an immune reaction) was a primary goal in our studies.

As the reviewer implies this comment, it is also possible that other changes in the transfected cells could cause immune responses. The ultimate test for this possibility will be in human adipocyte to human studies in the future. Nonetheless, in response to the reviewer we did perform additional experiments to investigate this possibility in our mouse studies. We did visualize the implants by microscopy and found very little

macrophage infiltration in the implant adipose depots derived from either the NTC adipocytes or the NRIP1KO adipocytes. No evidence of unexpected inflammation was evident. To further investigate, *Mcp1* expression was measured on the implants and no difference was found between NTC or NRIP1KO adipocyte implants (Extended data Fig. 3i). We also determined whether increased systemic inflammation could be detected by measuring plasma cytokines. As shown in Extended Figure 3l, positive control LPS treated mice displayed large increases in the plasma cytokines measured—IL-1 β , TNF α , IL-6, IL1-0 and MCP-1. In contrast, non-treated mice, and mice with NTC or NRIP1KO adipocyte implants (from which the preadipocytes had both been treated with Cas9/sgRNA) had relatively low levels of serum cytokines/chemokines at the end of the study (Extended data Fig. 3l). These new data show no evidence of increased immune responses due to the implantation of adipocytes from Cas9 treated preadipocytes, consistent with the lack of Cas9 protein in the genetically modified adipocytes that are implanted (Fig. 1b).

4) Fig.1 g,h: Expression of UCP1 in Nrip1 targeted cells is not sufficient to claim induction of “thermogenic responses”. As indicated by the authors, targeting NRP1 has the potential advantage of upregulate expression of many genes that have favourable metabolic effects, compared to upregulation of UCP1 only as described by Chih-Hao Wang et al. Functional in vitro tests for energy expenditure cells and expression of thermogenic genes (more than those shown in Extended data Fig.4, such as Dio2, ADRB3, Glut1, PGC1a,..) have to be performed.

Response: We fully agree with the reviewer, and therefore we performed gene expression profiles on a greater number of thermogenic and beige genes, genes involved in fatty acid oxidation, oxidative phosphorylation and fatty acid transport (new Fig. 2a-b), as well as spending considerable time conducting measurements of oxygen consumption in control and NRIP1KO adipocytes. In addition, in response to reviewer 3 comments (see below), we also performed these experiments with adipocytes derived from both male and female mice to characterize any differences in responses to NRIP1KO due to sex. As shown in new Fig. 2e-j, the results are clear and dramatic in demonstrating increased thermogenic responses in the NRIP1KO adipocytes. In Fig. 2e (male adipocytes) and Fig. 2h (female adipocytes), it is seen that no difference in respiration is observed between NTC and NRIP1KO adipocytes under basal, unstimulated conditions, and oligomycin markedly inhibits respiration in equal fashion in the two conditions. This is expected, as there is no activation of UCP1 unless cells are stimulated with adrenergic agonist to activate lipolysis and release fatty acids to promote UCP1 uncoupling. Fig. 2f (male) and Fig. 2i (female) show this to be the case, as NE addition to the adipocytes elicits a greater increase in oxygen consumption in the NRIP1KO adipocytes which have elevated UCP1 expression. Most importantly, oligomycin inhibition of respiration is much less in the NRIP1KO adipocytes, as expected for the uncoupled state. These data are quantified in Fig. 2g (male) and Fig. 2j (female), showing high statistical significance of these findings. Thus, the thermogenic gene expression profile reflects a strong thermogenic effect that is physiologically functional. To our knowledge, these are the first data to show this effect in primary adipocytes in response to NRIP1KO in vitro.

5) Targeted human primary adipose progenitors is a strong point of the manuscript. Targeted cells should be more characterized.

Response: Yes, we fully agree that this is a very useful exercise, and have done more comprehensive analysis of both the mouse and the human cells in this study. We also refer the reviewer to the PNAS paper in 2019 from the Corvera group on extensive characterization of the human progenitors and adipocyte types. Our new results related to the Reviewer's comments are presented in new Figure 4 panels a-h and new Figure 6 panels a-g, showing extensive RNA-seq data for the mouse adipocytes and the human adipocytes that were genetically modified by the Cas9/sgRNA RNPs in both the NTC and NRIP1KO conditions. Using a recently published computational program described and used in our revised manuscript, we show that the RNA-seq data analysis shows that NRIP1KO causes a decrease in the overall "whiteness" of the human cell population and an increase in overall "browning" of these cells. The other results of these new RNA-seq datasets are described in detail in the revised manuscript and will serve as useful resources for the field.

Minor point:

Figure 2f: Time of implant harvest is not indicated

Response: corrected

Reviewer 2:

Thus, the present study reveals the suitability of gene-engineered beige adipocyte cell therapy approaches for metabolic diseases. In fact, this is an emerging issue here and now (DOI: 10.1038/nrendo.2010.20): expanding the amount of active BAT and/or browning via transplantation holds promise as a therapeutic strategy for morbid obesity and diabetes. Several strategies have been pursued (BAT engraftment, brown adipose progenitor cell transplant), but very few genetic-modification experimental approaches have been reported to date (see refs 37, 45). Overall, the data presented in the manuscript is highly relevant and results are discussed appropriately in relation to the currently available literature. The reviewer has a few suggestions that could improve data presentation if implemented.

Response: We thank the reviewer for these favorable comments and that our approach holds promise as a therapeutic strategy for morbid obesity and diabetes. We also appreciate the comment that the data we present is highly relevant and the results are discussed appropriately in relation to current literature.

-Introduction. Indicate that NRIP1 is also denoted as RIP140.

Response: Done

-Fig 2. Characterization of implanted adipose tissue depots at study termination (16 weeks after implantation) should include other thermogenic/beige specific marker genes to see whether they also retained elevated expression as shown for UCP1. In the legend, the

number of biological replicates is confusing (c-e). -Fig 4. As above, the characterization of implanted adipose tissue is required at study termination (13 weeks after implantation of genetically-engineered human adipocytes).

Response: We agree this is an important point and thank the reviewer for raising it. Unfortunately, it is difficult to address this in detail, given the fact that some infiltration of endogenous mouse cells into the implants occurs and it is difficult to excise the implants without getting some of the surrounding control tissue. Therefore, analysis of the adipose depot implants is contaminated by these infiltrating cells and surrounding endogenous cells. Nonetheless, in response to the reviewer's point we have done this experiment and provide the result in new Fig. 4h. Here we performed RNA-seq on the NTC and NRIP1KO adipocytes prior to implantation and identified genes that were significantly elevated in response to NRIP1KO (a set of these are shown in Fig. 4h left panel). We then analyzed the expression of this set of upregulated genes in the implants following their excisions from mice at the end of the experiment. The idea was to determine whether these genes that are upregulated by NRIP1KO *in vitro* remain upregulated after many weeks of *in vivo* location. As shown in new Fig. 4h, lower panel, some of these genes do remain upregulated and they include *Ucp1*, *Fabp3* and *Cidea*, which are known to be among the most upregulated genes in response to NRIP1KO. This was very encouraging, and indicates the remarkable stability of the implants to remain thermogenic after many weeks in vivo. (We also see this in human implants but the noise is even higher with very high CT values in the RT-PCR results.) However, only a minority of the full set of analyzed, upregulated genes in vitro do remain detectably upregulated in vivo (Fig. 4h right panel). This is likely due to the decreased signal due to "dilution" of the RNA from the infiltrating and surrounding endogenous cells that are like control adipocytes and other cell types. A better analysis will be to use single cell RNA-seq in the future, but this is clearly beyond the scope of the present study.

-NSG mouse model. Mice were on HFD for 5 weeks but also were placed at thermoneutrality. Why? To increase HFD-induced obesity and/or susceptibility to the effects of knock-down NRIP1 on UCP1 induction?

Response: Yes, to increase HFD-induced obesity and glucose intolerance, as noted in the Corvera Nature Medicine paper, 2016.

-Authors should discuss why despite similar high efficiency of Nrip1 gene disruption at 7 loci by SpyCas9/sgRNA RNPs, that results in variable degrees of NRIP1 protein loss in murine primary adipocytes. Does it also occur in the screen of sgRNAs targeting different loci of NRIP1 in human adipocytes?

Response: This is an interesting phenomenon that we didn't anticipate, and has not been characterized in great detail. But we and experts in this field feel that the most logical explanation is that additional start sites for transcription and/or translation are able to skip over mutations in the DNA/RNA sequences that encode the N terminal regions of the protein. We state this in the revised manuscript in the Results section. As the reviewer requests, we have now performed the difficult IP/Western blot protocol with human cells (RIP140 antibodies are poor, so one has to start with large amounts of sample for IP and

use a different antibody for blotting). We were able to show that the human RIP140 protein is not decreased when an sgRNA is used that does not upregulate UCP1, while it is decreased when an sgRNA is used that does increase UCP1 expression. These new data on the human adipocytes are now included in new Fig. 5f, and indicates that the similar human and mouse gene structures are indicative of similar regulatory mechanisms that can skip beyond mutations in the areas encoding the N terminal regions of the gene.

-Methods. The protocol of primary mouse preadipocyte differentiation in culture should be detailed. At what day post-differentiation are the cells considered fully differentiated adipocytes? In the legend to Fig 1, analysis in in mature primary adipocytes on day 6 post differentiation means day 6 after induction of differentiation or 6 days after cells reached differentiation? Regarding electroporation of adipocytes, has the percentage of cell death due to the electroporation procedure been measured? Refs lack et als

Response: We have provided more details to these methods, as requested. With the detailed conditions that we describe for electroporation of the progenitor cells, we observe no evidence of significant cell death. Refs have been corrected.

Reviewer 3:

E Tsagkaraki et al. provide a powerful strategy for cell therapy in metabolic disease, consisting to deliver complexes of SpyCas9 protein and sgRNA ex-vivo targeting NRIP1 gene. The method is attractive but its application to human pathology requires further investigations and authors do not clearly explain how it could be envisageable.

Response: We thank the reviewer for agreeing that our strategy is powerful and for the comment that our text does not clearly explain how it could be envisageable. We have added much new text on this point, and we feel we have significantly improved such explanation. This comment is similar to other reviewers who point to the need to more fully explain the novelty. We have added much text to highlight the aspects of this work that are significant advances in a therapeutic strategy vs the research strategy that others have used. Please see the revised Abstract and the first paragraph of the Discussion, which we have also copied above in response to the first point of Reviewer 1.

1- The reasons why the authors choose to target NRIP1 are not very clear and need to be clarify in introduction.

Response: Thanks for this comment, and we have now devoted a full paragraph in the Introduction to why NRIP1 is an excellent target for enhancing browning of adipocytes related to its normal suppression of thermogenesis and secretion of beneficial factors that improve glucose homeostasis.

2- Conclusions appear sometimes not completely appropriate (Figure 2d) with results, in particular concerning the effects on glucose tolerance after NRIP1KO adipocytes implantation.

Response: We agree that our original manuscript had limited numbers of mice shown as the implantation experiments are tedious and long term. We have now greatly increased the number of implanted mice in our cohorts that we have now analyzed, and have added the new data into our revised manuscript. The original conclusions hold up nicely, but the statistical analyses provide even more confidence of these conclusions. We show statistical significance for improvement of GTT curves (Fig. 3f), and wish to also point out the high significance of the decreased fasting blood glucose values in the NRIP1KO adipocyte implanted mice (Fig. 3c).

3- What are the long-term consequences of high UCP1 expression in adipocytes? Are adipocytes still able to respond to acute stress?

Response: NRIP1KO adipocytes that express high UCP1 are very stable and continue to express high UCP1 for the duration of their time in culture. They continue to respond to NE, as shown in new Figure 2, and their responsiveness to stress inducers such as oligomycin and FCCP is normal. There does not appear to be deleterious effects of UCP1 expression in these adipocytes over the long term either, as we see healthy implanted adipocytes in vivo with continuous UCP1 expression over several months.

4- Authors should identify if side effects on the mouse aging (more than 45 old-weeks) could be observed

Response: We have not yet experimented with mice that are nearing 1 year of having implants. This will take many more months and we hope the reviewer will understand this data will not be available for quite some time.

5- Authors only used male mice. Due to the effects of NRIP1 on estrogen receptors, authors should perform same experiments with adipocytes from female mice ?

Response: Thanks for this comment, which is an important one. We feel one of the advantages of our system and approach is that the NRIP1KO is selective in the adipocytes since we transfect *ex vivo*, and therefore our procedures will not interfere with NRIP1 in other cell types, in particular reproductive organs. But your comment is a great one in order to see whether adipocytes from female mice respond in the same way as male adipocytes to NRIP1KO. We have therefore now performed extensive experiments in response to your comment and include this data on gene expression levels and thermogenesis in new Figure 2, panels a-j. Remarkably, you can see from panels a and b, the increases in gene expression of the most upregulated (*Ucp1*, *Cidea* and *Fab3*) as well as others that are less upregulated are all virtually identically regulated in both male and female adipocytes. This is also the case when we perform Western blots to check on UCP1 protein expression in the response to NRIP1KO—virtually identical upregulation in male versus female adipocytes (Fig. 2c).

In addition, we performed experiments on oxygen consumption with adipocytes derived from both male and female mice to characterize any differences in responses to NRIP1KO due to

sex (also related to Reviewer 1 comments). As shown in new Figure 2e-j, the results are clear and dramatic in demonstrating similar increased thermogenic responses in the NRIP1KO from both male and female adipocytes. In Fig. 2e (male adipocytes) and Fig. 2h (female adipocytes), it is seen that no difference in respiration is observed between NTC and NRIP1KO adipocytes under basal, unstimulated conditions, and oligomycin markedly inhibits respiration in equal fashion in the two conditions. This is expected, as there is no activation of UCP1 unless the cells are stimulated with adrenergic agonist to activate lipolysis and release fatty acids to promote UCP1 uncoupling. Fig. 2f (male) and fig. 2i (female) show this to be the case, as NE addition to the adipocytes elicits a greater increase in oxygen consumption in the NRIP1KO adipocytes which have elevated UCP1 expression. Most importantly, oligomycin inhibition of respiration is much less in the NRIP1KO adipocytes, as expected for the uncoupled state. These data are quantified in Fig. 2g (male) and Fig. 2j (female), showing high statistical significance of these findings. Thus, the thermogenic gene expression profile reflects a strong thermogenic effect that is physiologically functional and related to your comment, the same is true for both male and female adipocytes.

6- Why authors did not perform adipocytes implantation directly in WAT or BAT tissues ?

Response: With adipocyte implantations, unlike tissue implantations, we need to use Matrigel as the matrix for the cells as they are implanted. Matrigel is a liquid at room temperature and solidifies at physiological temperatures. Thus, it is more feasible to implant in a subcutaneous location near but not within endogenous adipose depots. The Corvera group has also shown that implanting mature adipocytes with Matrigel achieves vascularization after implanting into the subcutaneous space. We have utilized these exciting findings to expand on these original findings that implanting into a fat depot is not necessary for engraftment and function of the implanted cells to work as a metabolically active organ.

7- Concerning human subjects, what are the clinical characteristics :age, sex... ?

Response: The samples we use for human adipose tissue starting material are derived from fresh surgical material (see methods), and although consent is obtained all information except gender is de-identified. Thus we have no information about the human subjects. However, interestingly, all the human adipose samples used in this study are from female subjects.

Reviewer 4:

The paper is well written, and the experimental design and flow is OK.

The genome editing experiments are performed well and the SANGER sequencing followed by analysis of the mRNA and protein levels of NRIP1 is appropriate and conclusive.

Response: Thanks for these comments.

I would be interested in seeing additional off-target analysis using guide-seq or deep sequencing.

Response: We agree with the Reviewer that off target analysis using GUIDE-seq/deep seq would be more rigorous and extensive than the methods we initially used. We have therefore embarked on the most rigorous protocols for our new experiments to define off target effects. We now report that although there were some significant off target effects in the mouse system, we were excited to find that was not the case for the edited human adipocytes. Specifically, to address this important point, we used the genome wide, unbiased identification of DSBs enabled by the sequencing (GUIDE-seq) approach as described in Tsai et al., Nat Biotechnology, 2015 with some modifications. Briefly, to precisely mimic our implanted cells, we optimized the transfection to deliver dsODN with our RNP concentration, the same electroporation method and in the mouse and human primary preadipocytes instead of a cell line, achieving great on-target editing and dsODN integration. In the murine primary preadipocytes, several potential off-target sites were detected and top candidates (in total 11 loci with up to 5 sgRNA mismatches) were screened with deep sequencing. In total, about 5 out of these 11 loci were proved to be real sites of off-target editing but located in intronic or unannotated genomic regions. IMPORTANTLY, in the human primary preadipocytes, GUIDE-seq detected virtually no off-target sites. Therefore, we selected the top 5 candidates predicted by the Cas-offinder web tool with up to 4 sgRNA mismatches and based on the distance of the mismatches from the PAM to screen with amplicon deep sequencing. In agreement with our GUIDE-seq results, no off-target editing was detected in the human mature adipocytes. We now include this data in new Extended data Figure 4.

My most important concern has to do with the novelty of this approach and manuscript. More specifically, and I am including representative papers (not a complete list), recent results show direct tissue-grafting approach to increasing endogenous brown fat (PMID: 29785004) or through editing of lncRNAs (PMCID: PMC5899897) or through CRISPR-based editing of uncoupling protein 1 expression (PMID: 32848096). Moreover, the delivery of CRISPR to fat tissue is not novel (PMID: 31467027) nor is the observation that NRIP1 is important (shown by the same group, PMID: 30190322).

Response: Thank you for this important comment and the citations of previous work which we are aware of and have cited all of them in the manuscript. We understand the good reason for this criticism, based on the first version of the manuscript, which failed to adequately and fully describe the novelty of the study. This is similar to the comment made by Reviewer 1 and our reply is similar: This is a really important point that is critical to the work, and we thank the reviewer for noting that we have understated the novelty of our approach and results. While the papers cited are of keen interest, our work is instead devoted to advancing conditions of procuring and genetically modifying human adipocytes in the context of a therapeutic strategy. Previous work is not in this mode, as primary human cells from expanded progenitors were not used and methods of transfection were performed with vectors with continuous expression of immunogenic Cas9. As we now emphasize in the revised Abstract, Introduction and Discussion sections of this resubmitted manuscript, we report on methods developed and applied that can be more easily advanced to a therapeutic

stage. We now explicitly state 5 criteria we have used in our studies that emphasize the novelty of our findings towards therapeutic purpose. Thus, we now state in the Discussion section:

“A major goal of the present study was to advance the application of CRISPR technology to metabolic disease in the context of a potential therapeutic strategy. The starting point of our experimental plan was the success of many laboratories in demonstrating the efficacy of implanting mouse or human brown/beige adipocytes into glucose intolerant mice to alleviate diabetes^{21,23,40-42}. Five key criteria were incorporated into our approach: 1. Generation of large numbers of adipocyte progenitors from small samples of human adipose tissue, 2. Identification of a strong suppressor of adipocyte browning for targeting by CRISPR to optimize the therapeutic benefit of adipocyte implantation, 3. Stealth administration of SpyCas9/sgRNA to cells that would not expose recipient mice to immunogenic reagents, 4. Minimizing off target effects, and 5. High efficiency gene disruption in which most cells *ex vivo* are affected in a single step without a cell selection step. Taken together, the data presented here show that our methods using CRISPR-based RNPs targeting *Nrip1* to a large extent satisfy the above criteria. These methods can indeed enhance browning of mouse or human adipocytes *ex vivo* at high efficiency without the use of expression vectors to improve glucose homeostasis in obese mice.”

In addition to the above, we have added much new data to the original study with 3 additional full multi-panel Figures plus Extended data Figure 4, plus other panels in other Figures. This wealth of data on our system that we present in this revision, combined with the elements that can lead to further advancement towards extending the proof of concept we demonstrate here in mice, could in the future be applied to larger animals and potentially to the clinic, bringing novelty and importance beyond the published record. We hope the Reviewers agree.

While the authors attribute the relevance of upregulating NRIP1 to the resulting expression of many genes that have "favorable" metabolic effects, this is not clearly shown (with data) in the manuscript. Furthermore, I would argue that the comparison with UCP1 should be performed and shown in the manuscript. To conclude, considering the recent literature results, I don't believe the paper presents a sufficient advance to warrant publication in nature communications.

Response: Thanks for these comments, which also relate to the above points on the novelty of the study. We feel the reviewer's point was well taken based on the original version of the manuscript, which did not adequately highlight the novelty and did not clearly detail the significance of the scientific advances. We have now added much new data and major new text to address this problem. These new extensive results are included in 3 additional multi-panel Figures, in new panels within old Figures and in the new or revised Extended Figures 3 and 4 in the revised version. Please see the revised Abstract and first paragraph of the revised Discussion in the new version for such additional text in addition to much new text in red font throughout. There are no other published papers that use primary adipocytes that can be greatly expanded from small amounts of human adipose tissue and genetically modified by CRISPR to enhance their therapeutic functions *in vivo* in the absence of continued Cas9 protein expression. We feel our study establishes proof of concept for this therapeutic approach that moves well beyond previous work in the field.

REVIEWERS' COMMENTS

Reviewer #1 (Remarks to the Author):

The authors fully addressed my concerns.

But I have a new one linked to the sentence added in the abstract of the revised version that is:
"Here we applied methods to greatly expand human adipocyte progenitors from small samples of human subcutaneous adipose tissue"

There is no details in the Methods section or in the text to support this claim. This is of a great interest in the field and the authors should precise what is the method used

Reviewer #2 (Remarks to the Author):

The authors have carefully responded to questions and criticisms. New data and changes made by the authors upon referee's suggestions have significantly improved the interpretation and discussion of the data. I consider it as a relevant contribution to this field of study.

Reviewer #3 (Remarks to the Author):

The revisions done by the authors are appropriate.

Added new data further improve the manuscript.

Reviewer 1 stated: The authors fully addressed my concerns. But I have a new one linked to the sentence added in the abstract of the revised version that is: "Here we applied methods to greatly expand human adipocyte progenitors from small samples of human subcutaneous adipose tissue" There is no details in the Methods section or in the text to support this claim. This is of a great interest in the field and the authors should precise what is the method used.

Our response: We have now included in the Methods section a sentence stating that the method we used for this aspect of the study, provided in detail in reference 23, has been demonstrated to greatly expand human adipocyte progenitor cells from small samples of human adipose tissue. Reference 23 has all the details of this method which we have applied and combined with our new CRISPR methods in the present manuscript.